# Transition metal ion FRET uncovers $K^+$ regulation of a neurotransmitter/sodium symporter

Christian B. Billesbølle[1], Jonas S. Mortensen[1], Azmat Sohail[2], Solveig G. Schmidt[1], Lei Shi[3], Harald H. Sitte[2], Ulrik Gether[1] & Claus J. Loland[1]

Neurotransmitter/sodium symporters (NSSs) are responsible for $Na^+$-dependent reuptake of neurotransmitters and represent key targets for antidepressants and psychostimulants. LeuT, a prokaryotic NSS protein, constitutes a primary structural model for these transporters. Here we show that $K^+$ inhibits $Na^+$-dependent binding of substrate to LeuT, promotes an outward-closed/inward-facing conformation of the transporter and increases uptake. To assess $K^+$-induced conformational dynamics we measured fluorescence resonance energy transfer (FRET) between fluorescein site-specifically attached to inserted cysteines and $Ni^{2+}$ bound to engineered di-histidine motifs (transition metal ion FRET). The measurements supported $K^+$-induced closure of the transporter to the outside, which was counteracted by $Na^+$ and substrate. Promoting an outward-open conformation of LeuT by mutation abolished the $K^+$-effect. The $K^+$-effect depended on an intact Na1 site and mutating the Na2 site potentiated $K^+$ binding by facilitating transition to the inward-facing state. The data reveal an unrecognized ability of $K^+$ to regulate the LeuT transport cycle.

---

[1] Molecular Neuropharmacology and Genetics Laboratory, Department of Neuroscience and Pharmacology, Faculty of Health and Medical Sciences, University of Copenhagen, Copenhagen N 2200, Denmark. [2] Center for Physiology and Pharmacology, Institute of Pharmacology, Medical University Vienna, 1090 Vienna, Austria. [3] Computational Chemistry and Molecular Biophysics Unit, National Institute on Drug Abuse, NIH, Baltimore, Maryland 20852, USA. Correspondence and requests for materials should be addressed to C.J.L. (email: cllo@sund.ku.dk).

Neurotransmitter/sodium symporters (NSSs) play an essential role in terminating neurotransmitter action in the central nervous system[1] and operate by utilizing the energy stored in the transmembrane $Na^+$-gradient as driving force for substrate transport. Key members of the family include transporters of the neurotransmitters dopamine (DAT), norepinephrine and serotonin (SERT)[2,3]. As a consequence of NSSs involvement in controlling synaptic signaling, the transporters have been established as targets for many important therapeutics[4] and are targeted by illicit drugs such as cocaine, amphetamines[5] and cathinone-derivatives ('bath salts')[6]. In addition to co-transport of $Na^+$, all mammalian NSSs are dependent on co-transport of $Cl^-$ (ref. 4). Importantly, counter-transport of $K^+$ was reported to stimulate the rate of serotonin (5-HT) uptake by human SERT[7,8]. Furthermore, it was shown that $H^+$ can substitute for $K^+$ as counter-transported cation in SERT[9], and it was proposed that co-transport of $Cl^-$ or counter-transport of $H^+/K^+$ is a common feature of charge-balance in NSS[10]. However, to our knowledge, there is no evidence for $K^+$-counter-transport in other NSSs than SERT and the molecular details behind its function are largely unknown.

High-resolution X-ray crystal structures of the leucine transporter (LeuT) from Aquifex aeolicus[11], the multi-hydrophobic amino acid transporter from Bacillus halodurans[12], DAT from drosophila melanogaster[13,14] and the human SERT[15] have provided important insight into the molecular structure of this class of proteins. NSSs possess 11 or 12 transmembrane (TM) segments with a substrate-binding site and one or two adjacent $Na^+$-binding sites (Na1 and Na2) buried approximately half-way across the membrane bilayer[11,12,14,16]. The conventional model for ion-coupled substrate-transport has been inferred from crystal structures of 'canonical' transport-cycle intermediates (outward-facing, substrate-bound occluded and inward-facing)[11,17]. Indeed transport is widely assumed to be governed by a simple allosteric mechanism[18], where access to and from the substrate-binding site is tightly regulated by intra- and extracellular gating domains[19–23].

LeuT has been used as the principal model system for studying NSS structure–function relationships[16,24] and has served as template for homology models that have guided, for example, binding site mapping in mammalian NSSs[25–28]. In addition, dynamic inferences about conformational transitions in the NSS transport cycle have been obtained using LeuT as model in studies using biophysical techniques such as double electron–electron resonance[29–31], single molecule fluorescence resonance energy transfer (smFRET)[32,33] and site-directed fluorescence quenching spectroscopy[34]. Nonetheless, despite a growing mechanistic understanding of the translocation process, none of the studies have suggested a putative role of $K^+$ and thus addressed the question whether the counter-transport of $K^+$ proposed for SERT[8,35] is indicative of a general regulatory role of $K^+$ in other NSS proteins.

In the present study, we provide structural evidence for a role of $K^+$ in regulating conformational transitions in LeuT. We demonstrate that $K^+$ interacts with LeuT and competitively inhibits $Na^+$-dependent binding of substrate. Moreover, we find that internal $K^+$ stimulates [³H]alanine uptake by LeuT in proteoliposomes. Next, we use transition metal ion fluorescence resonance energy transfer (tmFRET), as a highly sensitive method for direct measurements of conformational changes in response to $K^+$ and show that $K^+$ inhibits $Na^+$ and substrate binding possibly by binding an outward-closed conformation of the transporter. The effect of $K^+$ requires an intact Na1 site and is abolished by mutation of Arg30 that promotes the outward-open conformation of LeuT. Finally, $K^+$ potency is increased upon mutational disruption of conserved interactions associated with the Na2 site thereby biasing the inward-open conformation.

Our data suggest altogether that $K^+$ could play a canonical role in regulating the function of NSS proteins. Specifically, we propose a model in which internal $K^+$ facilitates substrate turnover by inhibiting substrate rebinding in the outward-closed/inward-facing conformation. This will inhibit substrate efflux and result in an increased concentrative capacity of the transporter and, thus, substrate transport against a larger chemical gradient.

## Results

**$K^+$ competitively inhibits $Na^+$ binding to LeuT.** $K^+$ counter-transport by SERT was reported almost four decades ago[7,8]. However, it has not been addressed whether the role of $K^+$ in SERT is indicative of a general feature in the NSS transport mechanism. To address this question, we set out to investigate whether $K^+$ interacts with LeuT. We purified LeuT from Escherichia coli and confirmed binding of [³H]leucine in 200 mM $Na^+$ or $Li^+$ using the scintillation proximity assay (SPA) on detergent-solubilized protein as previously reported[36]. Note that all subsequent experiments are performed on LeuT in detergent (dodecyl-β-D-maltoside, DDM) unless otherwise stated. We observed no [³H]leucine binding in 200 mM $K^+$ or other tested monovalent cations (Fig. 1a). To assess the effect of $K^+$ on LeuT, we performed $Na^+$-dependent [³H]leucine binding where $Na^+$ was substituted with either $K^+$ or choline (Ch$^+$) to maintain a total ionic concentration of 200 mM. Remarkably, we observed a 3-fold increase of the $EC_{50}$ for $Na^+$ when substituted with $K^+$ compared with Ch$^+$ (Fig. 1b). Likewise, we observed a 3-fold increase in $EC_{50}$ for $Li^+$ substituted with $K^+$ compared with Ch$^+$ (Fig. 1b). We proceeded by investigating $Na^+$-dependence of [³H]leucine binding in the presence of fixed $K^+$ concentrations (Fig. 1c) and performed a Schild analysis of these data showing the shifts in $EC_{50}$ as a function of $[K^+]$. The Schild analysis provides information whether or not antagonism is competitive in nature: if the regression in a Schild plot is linear with a slope of 1, then the antagonism is competitive[37]. When competitive antagonism is observed, it also allows for determination of the affinity ($K_B$) of the antagonist. Here, the Schild plot revealed a linear correlation with a slope of 0.9 [0.8;1.0] (mean [95% confidence interval]) and an equilibrium constant ($K_B$) for $K^+$ of 176 [153;203] mM (Fig. 1d). This suggests that $K^+$ competitively inhibits $Na^+$ binding to LeuT WT.

To test whether $K^+$ interaction with LeuT affected transport, we reconstituted LeuT into proteoliposomes containing buffer with 200 mM KCl or CsCl. [³H]alanine uptake by the proteoliposomes was measured for up to 45 min in external buffer containing 200 mM NaCl. Interestingly, uptake was faster in $K^+$ compared with Cs$^+$ and reached a more than 2-fold higher maximum (Fig. 1e). Previously, $Na^+$/substrate symport-coupled $H^+$ antiport was suggested for prokaryotic NSS proteins[27,38] and for the homologous NSS protein Tyt1, where it was shown that substrate transport elicits $H^+$ efflux[38]. To test whether $H^+$ could substitute for $K^+$, proteoliposomes were formed in a CsCl buffer (200 mM) pH 6.5 or pH 8 and [³H]alanine uptake were assessed as in Fig. 1d at pH 8. However, the $H^+$ gradient appeared to have no effect on uptake: In proteoliposomes formed at pH 8, [³H]alanine uptake was 106±6% relative to uptake at pH 6.5 (means±s.e.m., $n=3$), suggesting that protons cannot substitute for $K^+$ as well as that a proton gradient does not stimulate uptake, at least not under the conditions used in the present investigation. Another possible interpretation of the result would be that internal $K^+$ facilitates transport by binding to LeuT in a way that inhibits rebinding of $Na^+$ and, accordingly, also substrate. This should reduce substrate efflux and result in an increased concentrative capacity

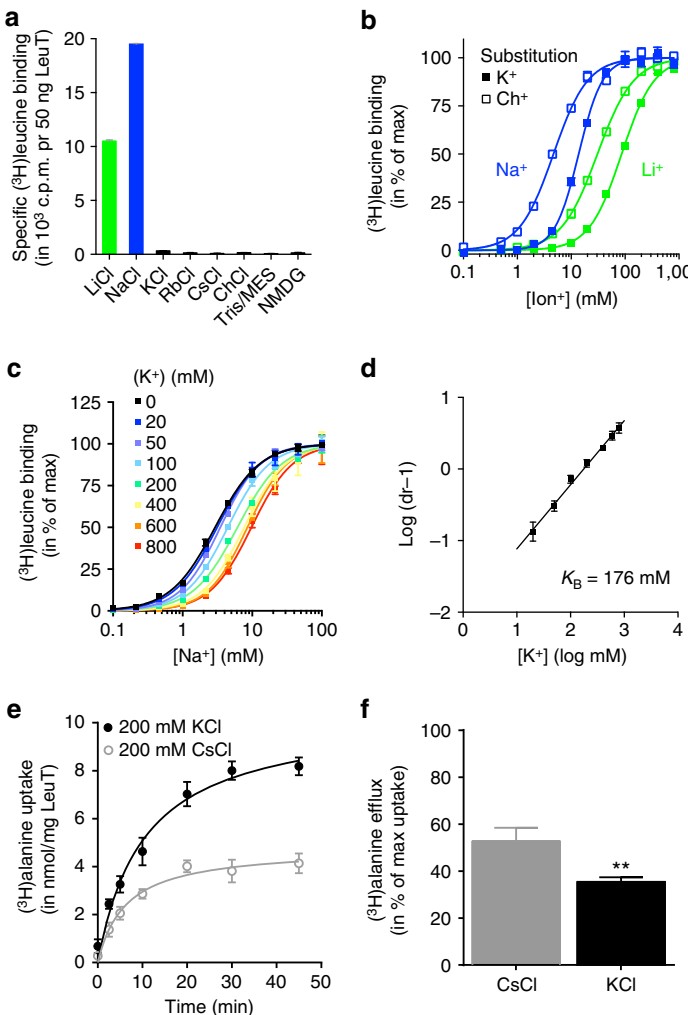

**Figure 1 | K$^+$ interacts with LeuT.** (**a–d**) Radioligand binding experiments suggests that K$^+$ inhibits substrate binding to LeuT. (**a**) Binding of 200 nM [$^3$H]leucine in 200 mM of the indicated salts to LeuT WT by scintillation proximity assay (SPA). Specific binding is observed in LiCl and NaCl. None of the other indicated salts promoted [$^3$H]leucine binding different from background ( + 5 mM alanine). (**b**) [$^3$H]leucine (1 μM) binding to LeuT WT is dependent on Na$^+$ or Li$^+$. Na$^+$ (blue symbols) was substituted with either Ch$^+$ (open squares, EC$_{50}$ = 4.68 [4.16;5.27] mM) or K$^+$ (filled squares, EC$_{50}$ = 13.8 [12.8;14.8] mM). Li$^+$ (green symbols) was substituted with either Ch$^+$ (open squares, EC$_{50}$ = 30.7 [28.2;33.4] mM) or K$^+$ (filled squares, EC$_{50}$ = 86.2 [80.9;91.8] mM). (**c**) Na$^+$-dependent [$^3$H]leucine (1 μM) binding to LeuT WT in the presence of the indicated K$^+$ concentrations. Na$^+$ is substituted with Ch$^+$. (**d**) Schild plot derived from data shown in **c** follows kinetics compatible with competitive inhibition of Na$^+$-dependent [$^3$H]leucine binding by K$^+$ (K$^+$ binding constant ($K_B$) of 176 [153;203] mM, $R^2$ = 0.97). (**e**) Reconstitution of LeuT into proteoliposomes in 200 mM of either K$^+$ (black circles) or Cs$^+$ (grey circles). [$^3$H]alanine (0.5 μM) uptake is performed in buffer with 200 mM Na$^+$. Specific [$^3$H]alanine uptake was ∼2-fold higher for proteoliposomes with internal K$^+$ relative to Cs$^+$ (10.4 [9.00;11.7] and 4.80 [4.08;5.52] nmol mg$^{-1}$ LeuT, respectively). Proteoliposomes containing Na$^+$ (no gradient) was used to determine nonspecific [$^3$H]alanine activity. Data points are means ± s.e.m.; $n$ = 3. (**f**) Efflux experiments showing the relative amount of [$^3$H]alanine released from proteoliposomes over 30 min with K$^+$ or Cs$^+$ (200 mM) on the inside causing an efflux of 35 ± 2% and 53 ± 6%, respectively (means ± s.e.m., $n$ = 6). Data are given as % of initial amount of [$^3$H]alanine in the proteoliposomes. Proteoliposomes were preincubated with [$^3$H]alanine (50 nM) for 10 min and efflux measurements were initiated by a 5 × dilution into a Na$^+$ buffer containing a high concentration (500 nM) of unlabelled alanine to prevent any reuptake of released [$^3$H]alanine. **$P$ < 0.01, unpaired $t$-test.

of the transporter, that is, substrate can be transported against a larger chemical gradient. To address this question, we performed an efflux experiment. Proteoliposomes were preincubated with [$^3$H]alanine (50 nM) for 10 min and efflux measurements were initiated by diluting the proteoliposomes five times into a Na$^+$ buffer with a high concentration (500 nM) of unlabelled alanine to prevent any reuptake of released [$^3$H]alanine. The amount of [$^3$H]alanine efflux was assessed by comparing the remaining internal [$^3$H]alanine after 30 min to the initial amount. In agreement with this hypothesis, [$^3$H]alanine efflux was significantly lower for K$^+$-containing proteoliposomes compared with control (Fig. 1f).

**Application of tmFRET in LeuT.** To have means for direct investigation of the putative effect of K$^+$ on LeuT conformation, we implemented tmFRET, which is a powerful tool for direct assessment of changes in intramolecular distances on, for example, transitions between conformational states. The method takes advantage of the ability of transition metals to act as a non-fluorescent FRET acceptor by quenching the emitted fluorescence from a nearby fluorophore in a distance-dependent manner[39,40] (Fig. 2a–c and Supplementary Fig. 1a–e). Specifically, fluorescence from a Cys-conjugated fluorophore (fluorescein, FL) was quenched by a coloured transition metal ion (for example, Ni$^{2+}$), chelated in an engineered metal-binding site (for example,

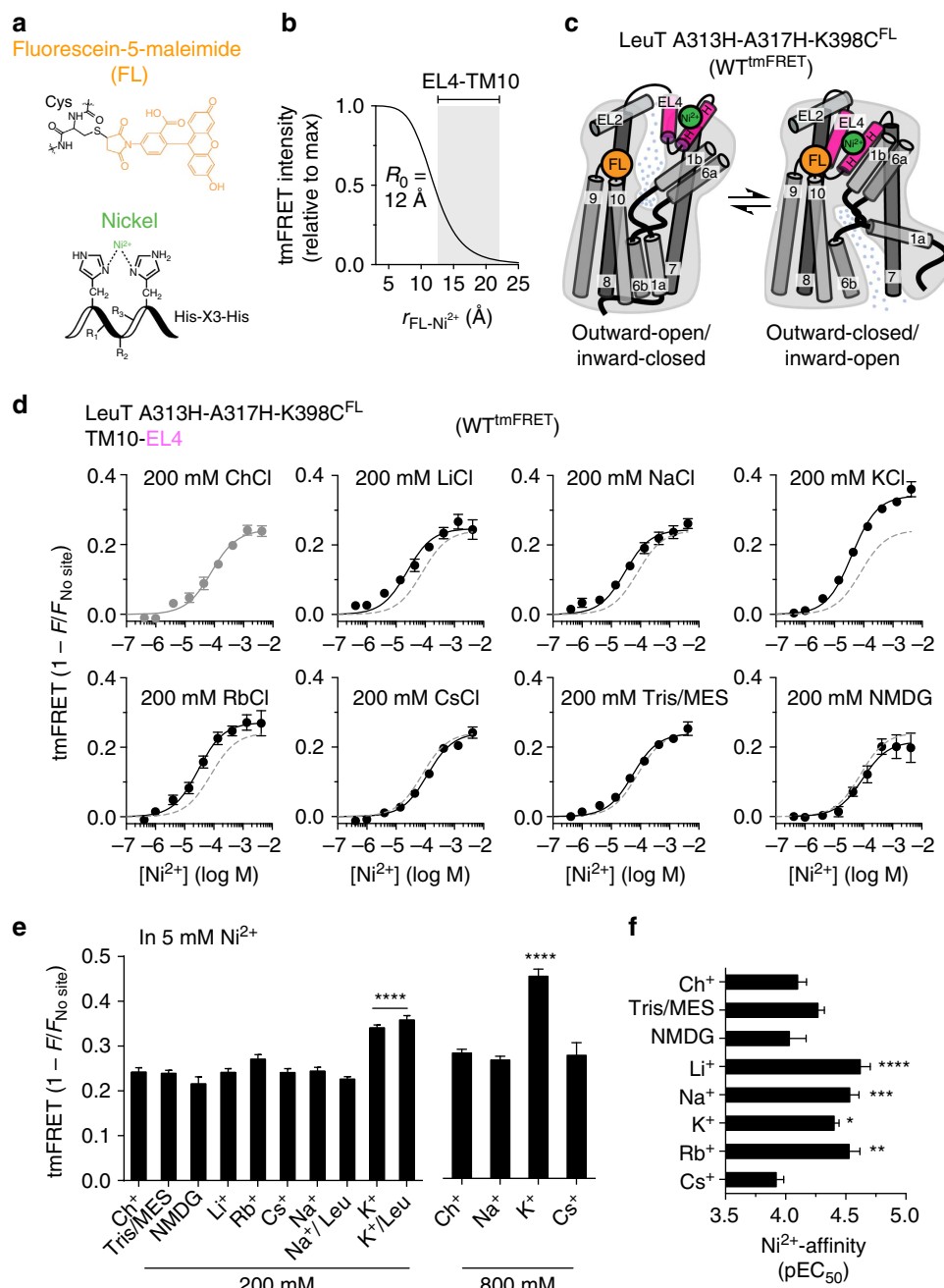

**Figure 2 | K$^+$ induces a different tmFRET state than other tested cations.** (**a**) Chemical structures of fluorescein-5-maleimide (orange) conjugated to a cysteine, and a nickel ion (green) coordinated by the imidazole moieties of two histidines inserted in an alpha-helical motif one helix-turn apart (His-$X_3$-His). (**b**) Expected dynamic range (grey area) of the EL4:TM10 tmFRET pair-based LeuT-crystal structures and the FL-Ni$^{2+}$ Förster distance ($R_0 = 12$ Å). (**c**) Cartoon of LeuT with EL4:TM10 tmFRET pair shown in outward open and outward closed states depicting the principle behind changes in tmFRET as a result of conformational changes. The tmFRET pair reports distance-dependent quenching of fluorescence from FL conjugated to an inserted cysteine at the top of TM10 (orange sphere, K398C) by Ni$^{2+}$ coordinated by a His-$X_3$-His motif in EL4 (green sphere, A313H-A317H). (**d–f**) Detecting distance–dependent fluorescence quenching as a function of Ni$^{2+}$ in LeuT with EL4:TM10 tmFRET pair (LeuT A313H-A317H-K398C$^{FL}$; see Supplementary Fig. 1 for tmFRET proof-of-principle applied to LeuT). (**d**) tmFRET intensity as a function of [Ni$^{2+}$] performed in 200 mM of the indicated salts yielded saturable tmFRET response. Dotted line represents tmFRET in ChCl (grey) for comparison. (**e**) Maximal tmFRET values from the experiments performed in **d** compared with experiments performed in 800 mM of the indicated salts. K$^+$ significantly and dose-dependently increases tmFRET values, consistent with a decrease in the TM10-EL4 mean distance. The addition of leucine (100 μM) to Na$^+$ or K$^+$ does not change tmFRET response relative to the ion alone. (**f**) Ni$^{2+}$ affinities (pEC$_{50}$) obtained from experiments shown in **d**, indicate that Na$^+$, Li$^+$, K$^+$ and Rb$^+$ (compared with Ch$^+$) induce a change in the EL4 secondary structure that stabilizes a configuration with increased Ni$^{2+}$ affinity. Data points are means ± s.e.m., $n = 3$–6. *$P < 0.05$; **$P < 0.01$; ***$P < 0.0005$; ****$P < 0.0001$; denote significance level from a one-way analysis of variance with *post hoc* Bonferroni's multiple comparison test relative to the Ch$^+$ condition in the same experimental setup.

His-$X_3$-His of an α-helix, Fig. 2a and Supplementary Fig. 1a–e). Importantly, tmFRET between FL and $Ni^{2+}$ occurs at a short range and is highly distance dependent, having a Förster distance ($R_0$) of ∼12 Å (refs 39–42; Fig. 2b). Accordingly, tmFRET provides an ideal sensitivity range (∼8–18 Å) for detecting changes in intramolecular distances predicted to take place in a transport protein like LeuT during transport (Fig. 2c)[16]. See the 'Methods' section and Supplementary Fig. 1 for a detailed implementation of tmFRET in LeuT.

**$K^+$ binding induces closure of extracellular domains.** Comparing the crystal structures of LeuT in the outward-open conformation ($Na^+$-bound, PDB 3TT1; ref. 17) with the outward-occluded ($Na^+$ and leucine bound, PDB 2A65; ref. 11) and inward-open (apo, PDB 3TT3; ref. 17) states

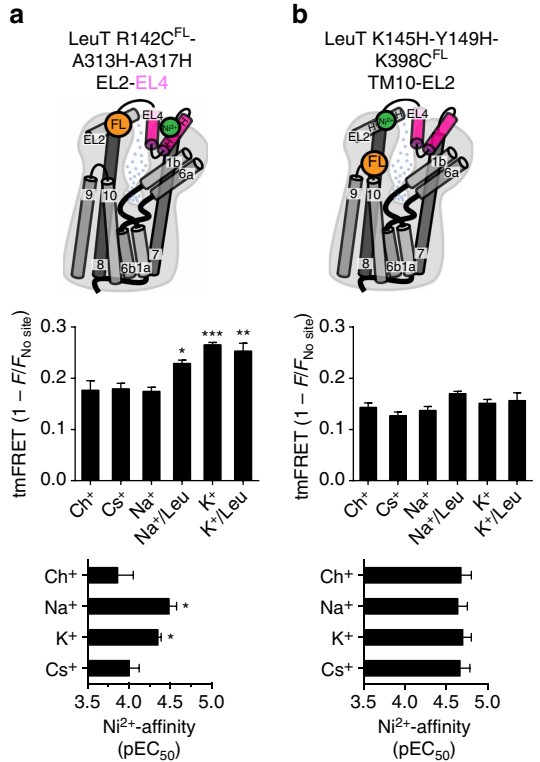

**Figure 3 | Further evaluation of EL4 movements by additional tmFRET probes.** (**a**) Top panel: illustrating tmFRET principle between EL2:EL4 with FL conjugated to an inserted cysteine in EL2 (orange sphere, R142C) and $Ni^{2+}$ coordinated by His-$X_3$-His motif in EL4 (green sphere, A313H-A317H). Middle panel: tmFRET results from LeuT R142C$^{FL}$-A313H-A317H in 5 mM $Ni^{2+}$ and 200 mM of the indicated ion. tmFRET is significantly increased by $K^+$ and $Na^+$/Leu suggesting a reduction in the mean distance between EL2 and EL4. As for the EL4:TM10 probing in Fig. 2, $Na^+$ shows no significant change compared with $Ch^+$ or $Cs^+$. The further addition of leucine (100 μM) does significantly increase tmFRET intensity. Bottom panel: $Ni^{2+}$ affinities (pEC$_{50}$) derived from tmFRET experiments on LeuT R142C$^{FL}$-A313H-A317H showing higher $Ni^{2+}$ affinity in $Na^+$ and $K^+$. (**b**) Top panel: tmFRET between principle TM10 and EL2 (LeuT K145H-Y149H-K398C$^{FL}$) as in **a** with results (middle panel) in 5 mM $Ni^{2+}$ and 200 mM of the indicated salts. tmFRET is not significantly affected by $K^+$, suggesting that the mean distance between TM10 and EL2 is not changed compared with application of the other indicated cations. Data points are means ± s.e.m., $n = 3$–6. *$P < 0.05$; **$P < 0.01$; ***$P < 0.0005$; denote significance level from a one-way analysis of variance with post hoc Bonferroni's multiple comparison test relative to the $Ch^+$ condition for the respective tmFRET pairs.

reveals a major shift in the positioning of extracellular loop 4 (EL4) that acts like a lid sealing the extracellular gate[11,17]. As a consequence, EL4 moves closer to the top of TM10 and to EL2 (Fig. 2c). To probe for conformational changes on the extracellular face of LeuT, induced by ion- and substrate binding, we generated a tmFRET pair by inserting a His-$X_3$-His site on EL4 and related its movement to FL conjugated to the top of TM10 (A313H-A317H-R398C, Fig. 2c–f). The purified and FL-labelled LeuT tmFRET variant retained WT-like [$^3$H]leucine binding (Supplementary Table 1 and Supplementary Fig. 2a–d). Also, the LeuT variant showed no significant change in the EC$_{50}$ for $Na^+$-stimulated binding of [$^3$H]leucine, when measured by $K^+$ substitution, indicating that interaction with $K^+$ was unaffected in the generated tmFRET pair (Supplementary Table 1 and Supplementary Fig. 2e). We proceeded by investigating the effect of cations on tmFRET between TM10 and EL4 (LeuT A313H-A317H-K398C$^{FL}$). The experiments were performed by increasing the concentration of $Ni^{2+}$ in buffers containing 200 mM of cations with varying ionic radii ($Li^+$; $Na^+$; $K^+$; $Rb^+$; $Cs^+$; Choline ($Ch^+$); $N$-methyl-D-glucamine (NMDG$^+$); tris(hydroxymethyl)aminomethane (Tris$^+$); Fig. 2d). Consistent with $Ni^{2+}$ quenching of FL bound to K398C, $Ni^{2+}$ caused a dose-dependent increase in the tmFRET value ($1 - F/F_{no\ site}$) for all cations used (Fig. 2d). Note that, by plotting $1 - F/F_{no\ site}$, we correct for dilution, collisional quenching and the inner-filter effect of $Ni^{2+}$ as the fluorescence from the tmFRET construct ($F$) is normalized to the fluorescence from the control construct devoid of the His-$X_3$-His site ($F_{no\ site}$; see also Supplementary Fig. 1). Remarkably, $K^+$ yielded a significantly higher maximum tmFRET value compared with all other cations used including both $Ch^+$ and $Na^+$ (Fig. 2d,e and Supplementary Table 2). This suggests that $K^+$, but not any of the other ions tested, promotes a decrease in the distance between TM10 and EL4. We next repeated the experiment in 800 mM of the cations, and again observed a substantially higher tmFRET in $K^+$ as compared with $Ch^+$, $Cs^+$ and $Na^+$ (Fig. 2e).

A feature of tmFRET is that the $Ni^{2+}$ affinity measured by titrating the quenching response may report transitions in the secondary structure supporting the His-$X_3$-His site[40,43]. Interestingly, the apparent affinities for the $Ni^{2+}$ response were significantly increased in $Li^+$, $Na^+$, $Rb^+$ and $K^+$, as compared with the large ions ($Ch^+$, $Cs^+$, NMDG$^+$ and Tris/MES; reflected in increased pEC$_{50}$ values, Fig. 2f). The shift in the pEC$_{50}$ values likely reports binding of $Li^+$, $Na^+$, $K^+$ and $Rb^+$ to LeuT; however, the effect must be uncoupled from the putative distance changes between EL4 and TM10 as there was no significant difference in $Ni^{2+}$ affinity between the application of $K^+$, and $Na^+$ and $Rb^+$.

To further evaluate possible movements of EL4, we investigated tmFRET between EL2 and EL4 (LeuT R142C$^{FL}$-A313H-A317H, Fig. 3a). Corroborating the observations for LeuT A313H-A317H-K398C$^{FL}$, we observed again significantly higher tmFRET in 200 mM $K^+$ compared with $Ch^+$, whereas the tmFRET signal was not significantly different in 200 mM $Na^+$ or $Cs^+$ compared with $Ch^+$ (Fig. 3a and Supplementary Table 2). Here, the application of leucine (in $Na^+$) did change the tmFRET signal significantly, suggesting a conformational transition in response to leucine binding between EL2 and EL4. The pEC$_{50}$ for $Ni^{2+}$ was for this construct also significantly increased in $Na^+$ and $K^+$ as compared with $Ch^+$ and $Cs^+$ (Fig. 3a, bottom panel). Finally, we generated a third tmFRET pair between two proposed static domains for reference (TM10-EL2, LeuT Y145H-K149H-K398C, Fig. 3b)[17,29,44]. As expected, we observed no significant differences in tmFRET for the different tested ions (Fig. 3b and Supplementary Table 2). Also, we observed no significant differences in the pEC$_{50}$ values for $Ni^{2+}$ (Fig. 3b,

bottom panel). Similar to the EL4-TM10 tmFRET pair, the EL2-EL4 and TM10-EL2 variants retained WT-like [³H]leucine binding and showed no significant change in the $EC_{50}$ for $Na^+$-stimulated binding of [³H]leucine (Supplementary Table 1 and Supplementary Fig. 2c–e).

Taken together, the tmFRET measurements corroborate our observations from radiotracer binding by suggesting that $K^+$ can interact with LeuT. The data also suggest that binding of $K^+$ promotes a movement of EL4 towards both TM10 and EL2. Notably, tmFRET between EL4 and TM10 measured in high $K^+$ (800 mM) approximately corresponded to the expected tmFRET value, calculated from distances between coordinates in the inward-facing crystal structure (Fig. 2e and Supplementary Fig. 3). In contrast, we observed no effect of $Na^+$ on tmFRET between EL4 and TM10, relative to that seen in the presence of non-binding cations.

**$Na^+$ competitively inhibits the conformational response to $K^+$.** To obtain a dose-response curve for the $K^+$-induced change seen in A313H-A317H-K398C$^{FL}$, which measures tmFRET between TM10 and EL4 and from now and onwards will be named WT$^{tmFRET}$, we added increasing $K^+$ concentrations (substitution with $Ch^+$) in the presence of a saturating (750 μM) $Ni^{2+}$ concentration (Fig. 4). $K^+$ produced ~22% increase in tmFRET with an $EC_{50}$ for $K^+$ of ~260 mM (Fig. 4a and Table 1). Note that these numbers are only rough estimates because we were unable to obtain full saturation owing to limitations in maximum attainable $K^+$ concentration in the assay (800 mM). The $K^+$-response was specific as no increase in FL quenching was observed when the experiment was carried out on LeuT K398C$^{FL}$ lacking the $Ni^{2+}$ site, or when $Ni^2$ was replaced with $Zn^{2+}$ (Supplementary Fig. 4a,b). In accordance with the experiments in Fig. 2, increasing concentration of $Na^+$ (again substitution with $Ch^+$) produced no conformational response in WT$^{tmFRET}$ (Fig. 4a). For LeuT R142C$^{FL}$-A313H-A317H, measuring tmFRET between EL2 and EL4, we also observed increasing tmFRET values in response to increasing $K^+$ concentrations, reaching a ~16% increase in the presence of 800 mM (Supplementary Fig. 4c). For LeuT K145H-Y149H-K398C$^{FL}$, measuring tmFRET between EL2 and TM10, we observed no response (Supplementary Fig. 4d).

We proceeded by investigating the ability of $Na^+$ and leucine to modulate the $K^+$-induced tmFRET response between TM10 and EL4 (WT$^{tmFRET}$ construct). Notably, the addition of 100 mM $Na^+$ markedly shifted the dose-response curve to the right resulting in an estimated increase in $EC_{50}$ for $K^+$ by ~3-fold (Fig. 4b). Thus, even though $Na^+$ had no effect on tmFRET by itself, it was able to right-shift the $K^+$-dose-response curve, suggesting that $Na^+$ is able to inhibit the $K^+$-induced conformational change detected by tmFRET. Further addition of leucine (50 μM) completely blocked the $K^+$-response (Fig. 4b). The effect by leucine, however, was $Na^+$-dependent since substitution with 100 mM $Cs^+$ (which does not support leucine binding, Fig. 1a) completely removed the blockade (Fig. 4b).

To further investigate the ability of $Na^+$ to compete for the $K^+$-bound conformation, we performed a tmFRET 'back titration' experiment (Fig. 4c). By keeping $K^+$ constant (600 mM), $Na^+$ yielded a FL de-quenching response that was fitted by a variable slope function with an $EC_{50}$ for $Na^+$ of 94.0 [80.9;109] mM (mean [s.e.m. interval]) and a $n_{Hill}$ of 1.13 ± 0.08 (mean ± s.e.m.), suggesting competition of $Na^+$ with one binding site (Fig. 4d and Table 1). Further addition of leucine dose-dependently potentiated the ability of $Na^+$ to compete the $K^+$-bound conformation with a 25-fold decrease in $EC_{50}$ for $Na^+$ in the presence of 50 μM leucine (Fig. 4d and Table 1). The addition of 50 μM alanine, a lower affinity ligand, which is

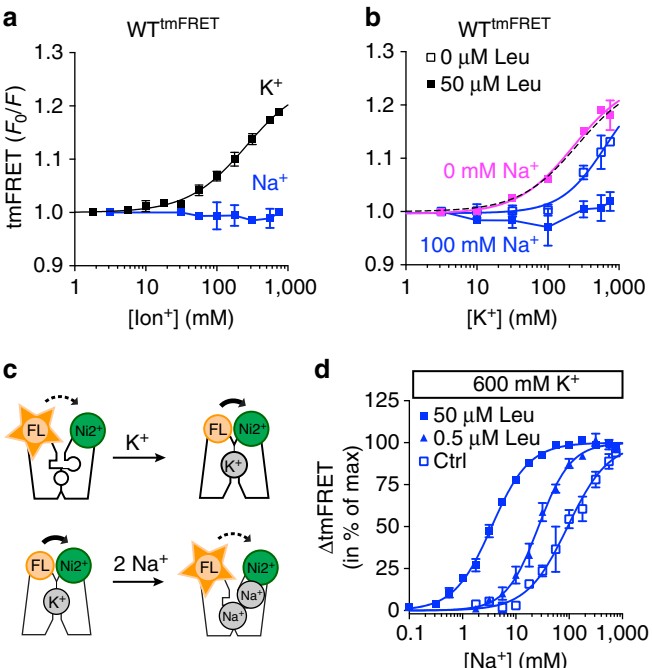

**Figure 4 | $K^+$ induced conformation competes with the coupled binding conformation of $Na^+$ and leucine.** (**a**) tmFRET between EL4 and TM10 as a function of $K^+$ (black) and $Na^+$ (blue) concentrations. $K^+$ produces a tmFRET-specific increase that fitted by the Hill equation indicates a single binding site ($n_{Hill} = 1.06 ± 0.15$) having an $EC_{50} = 260$ [137;495] mM. $Na^+$ showed no conformational effect. (**b**) The conformational effect by $K^+$ is attenuated by $Na^+$ and further blocked with leucine. Addition of $Na^+$ (100 mM) right-shifted the $K^+$ generated tmFRET response (blue, open squares) relative to $K^+$ alone from **a** (dotted line). Further addition of leucine (50 μM) completely inhibited the $K^+$ response (blue closed squares). The effect by leucine was $Na^+$-dependent since substitution with 100 mM $Cs^+$ completely removed the blockade (filled pink squares). Note that $Cs^+$ does not support binding of leucine. (**c**) Reaction scheme for tmFRET change in response to ions, $Na^+$ binding may be assayed by competitively inhibiting $K^+$-induced conformational change. Dotted arrow indicates low tmFRET, that is, high fluorescence; solid arrow indicates high tmFRET. (**d**) Normalized change in tmFRET between EL4 and TM10 ($F_0/F$) as a function of [$Na^+$]. LeuT A313H-A317H-K398C$^{FL}$ (WT$^{tmFRET}$) is stabilized in the $K^+$-bound conformation with 600 mM $K^+$ and 'back titrated' with $Na^+$, in the presence of 0, 0.5 or 50 μM Leu, which results in a $EC_{50}$ (in mM) for $Na^+$ of 94.0 [80.9;109], 26.6 [23.9;29.6] and 3.52 [3.29;3.76], respectively. $Ch^+$ was substituted for $Na^+$ or $K^+$ to maintain ionic strength. All the data are performed in 750 μM $Ni^{2+}$. Data are means ± s.e.m. $EC_{50}$ is given as mean with 95% confidence interval, $n = 3–8$.

also a better substrate for transport[34], yielded an $EC_{50}$ for $Na^+$ of 23.0 [21.4;24.7] mM (Supplementary Fig. 4e and Table 1). In contrast, $Li^+$ was less potent against $K^+$ ($EC_{50} > 400$ mM), and the addition of 50 μM leucine or alanine resulted in more modest potentiation of the $Li^+$ response (Supplementary Fig. 4f and Table 1). Taken together, our data suggest that $K^+$ promotes a more outward-closed state in LeuT that is inhibited by the coupled binding of $Na^+$ and substrate.

**LeuT in the outward-open conformation blocks $K^+$ binding.** Disruption of a conserved salt bridge between Arg30 (TM1) and Asp404 (TM10), located in the extracellular permeation pathway, has previously been shown to trap LeuT in an outward facing conformation[17,32]. We hypothesized that if $K^+$ preferentially

binds an outward-closed conformation, then disruption of the Arg30–Asp404 salt bridge should inhibit the $K^+$ response in LeuT. To disrupt the salt bridge, we mutated Arg30 to Ala in $WT^{tmFRET}$, followed by purification and fluorescent labelling of the resulting construct ($R30A^{tmFRET}$). Importantly, and further supporting that our tmFRET measurements most likely detects opening/closure of the transporter to the outside, the tmFRET measurements on $R30A^{tmFRET}$ showed a significant reduction in tmFRET in its apo-form ($Cs^+$) compared with $WT^{tmFRET}$. This suggests an increased distance between TM10 and EL4, in agreement with a more outward open conformation of the mutated transporter (Fig. 5a and Supplementary Fig 3d). The tmFRET was slightly increased in $Na^+$ and increased to a greater extend by further addition of leucine. However, $K^+$ induced no significant change in tmFRET relative to $Cs^+$ (Fig. 5a and Supplementary Table 2). To estimate the effect of R30A on $K^+$ potency, we performed a Schild analysis, which determined that the effect of $K^+$ ($K_B$) on $Na^+$-dependent [3H]leucine binding was essentially lost (Fig. 5b,c). Thus, disrupting the Arg30–Asp404 salt bridge abolishes the effect of $K^+$ on LeuT, presumably by biasing LeuT towards the outward-open state. This observation lends further support to the assumption that $K^+$ binding to LeuT involves a conformation that is different from the $Na^+$ and substrate bound conformation.

**The $K^+$-bound conformation requires an intact Na1 site**. Next, we assessed the consequence of mutations in the two

$Na^+$-binding sites on the $K^+$ effect. To investigate the role of the Na1 site, we mutated Thr254 to Val (T254V) in the $WT^{tmFRET}$ background ($T254V^{tmFRET}$). According to the high-resolution structures, Thr254 is one of the key coordinating residues of $Na^+$ in the Na1 site[11,17]. As would be expected, the $K_d$ for [3H]leucine binding (in 800 mM $Na^+$) was increased by over three orders of magnitude as Na1 is required for coordination of the substrate carboxy moiety (Fig. 6a and Supplementary Table 1). Unfortunately, this poor leucine affinity prevented a reliable Schild analysis of the $K^+$-effect on [3H]leucine binding. However, tmFRET measurements were possible. 800 mM of $Ch^+$ or $Cs^+$ yielded similar tmFRET values as observed for the background construct $WT^{tmFRET}$ (Fig. 6b and Supplementary Table 2), suggesting that the introduction of the T254V mutation did not change the distance measured between EL4 and TM10 (Fig. 6b). Furthermore, the substitution with 800 mM $Na^+$ or the addition of 100 μM leucine (with $Na^+$) did not significantly change the tmFRET values compared with $Ch^+$ (Fig. 6b). In contrast, 800 mM $K^+$ produced a significant increase in tmFRET; however, the magnitude of the tmFRET change was significantly reduced compared with $WT^{tmFRET}$. Thus, the T254V mutation impairs the $K^+$-response, which could be the result of either a disruption of $K^+$-coordination or a destabilization of the $K^+$-bound conformation.

**Na2 site disruption promotes an inward-facing conformation**. To assess role of the Na2 site, Thr354 was mutated to valine (T354V) in $WT^{tmFRET}$ ($T354V^{tmFRET}$). The T354V mutation was previously shown to abolish Na2 binding but not substrate binding in LeuT[17]. $T354V^{tmFRET}$ had reduced affinity for [3H]leucine ($K_d$ 5.45 ± 0.40 μM, in 800 mM $Na^+$) but the decrease was not nearly as large as for T254V (Fig. 7a). Strikingly, tmFRET values were dramatically increased for $T354V^{tmFRET}$ compared with $WT^{tmFRET}$ in buffers containing 800 mM of $Ch^+$ or $Cs^+$ (Fig. 7b and Supplementary Table 2), suggesting that $T354V^{tmFRET}$ is biased towards the outward-closed conformation similar to what we observe in 800 mM $K^+$ for $WT^{tmFRET}$ (Fig. 2). $Na^+$ (800 mM) did not significantly alter the tmFRET effect compared with $Ch^+$. However, the addition of 100 μM leucine (in 800 mM $Na^+$) significantly reduced the tmFRET value, whereas tmFRET intensity was unaffected by $K^+$ (Fig. 7b). These data suggest that mutation of the Na2 site

| Table 1 | Ion binding constants. | |
| --- | --- | --- |
| | EC$_{50}$ (mM) | $n_{Hill}$ |
| $Na^+$ | 94.0 [80.9;109] | 1.13 ± 0.08 |
| 500 nM Leu | 26.6 [23.9;29.6] | 1.42 ± 0.09 |
| 50 μM Leu | 3.52 [3.29;3.76] | 1.23 ± 0.05 |
| 50 μM Ala | 23.0 [21.4;24.7] | 1.35 ± 0.12 |
| $Li^+$ | 497 [411;600] | ND |
| 50 μM Leu | 22.5 [20.1;25.2] | 1.21 ± 0.07 |
| 50 μM Ala | 174 [163;187] | 1.58 ± 0.08 |
| $K^+$ | 260 [137;495] | 1.06 ± 0.15 |
| (+ T354D) | 198 [131;299] | 1.01 ± 0.10 |

Values were obtained experimentally by tmFRET measurements with LeuT A313H-A317H-K398C$^{FL}$ ($WT^{tmFRET}$). Data are shown as means ± s.e.m. or [s.e.m. interval], $n = 3-4$.

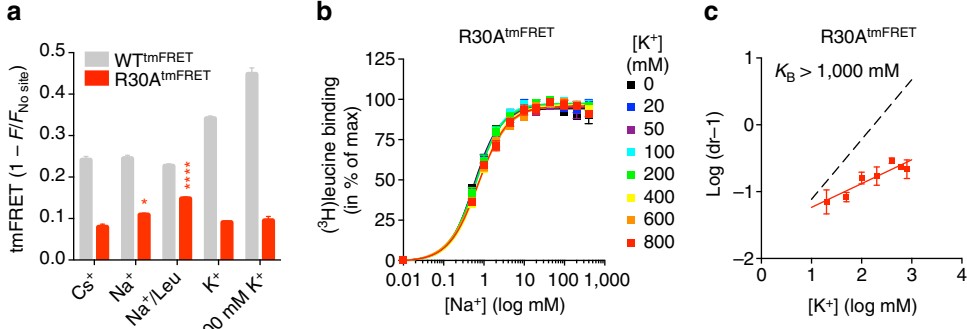

**Figure 5 | $K^+$-binding is abolished in LeuT R30A.** (**a**) tmFRET values between EL4 and TM10 obtained by tmFRET experiments with $R30A^{tmFRET}$ are consistent with an outward-open state for the $Cs^+$ condition (0.079 ± 0.007; means ± s.e.m., $n = 3$). $Na^+$ alone and together with leucine significantly increased tmFRET (0.108 ± 0.004 and 0.148 ± 0.004; means ± s.e.m., $n = 4$ and 7, respectively). No significant change in quenching was detected in the presence of $K^+$. $Ni^{2+}$ (5 mM) experiments were carried out with 200 mM of the indicated ions and 100 μM leucine ($Na^+$/Leu condition). *$P < 0.05$; ****$P < 0.0001$; denote significance level from a one-way analysis of variance with post hoc Bonferroni's multiple test relative to the $Cs^+$ condition. (**b**) $Na^+$ dependence of [3H]leucine binding to $R30A^{tmFRET}$ in the presence of the indicated $K^+$ concentrations. (**c**) Schild plot of the data shown in **b** show that $K^+$-inhibition of [3H]leucine binding to $R30A^{tmFRET}$ is severely compromised ($K_B > 1,000$ mM) with no significant effect of up to 800 mM $K^+$. Data points are means ± s.e.m., $n = 3-7$.

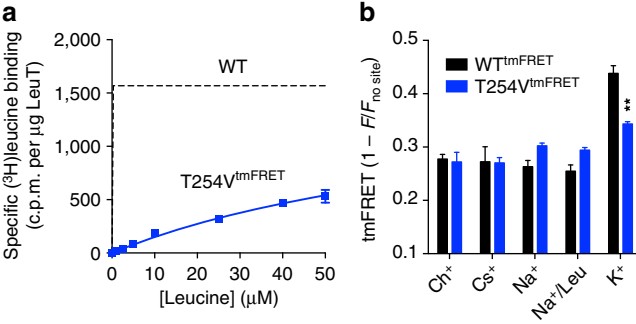

**Figure 6 | The K$^+$-induced conformation requires an intact Na1 site.**
(**a**) [$^3$H]leucine saturation binding for T254V$^{tmFRET}$ (LeuT T254V-A313H-A317H-K398C$^{FL}$; $K_D = 95.0 \pm 2.7\,\mu M$) performed in the presence of 800 mM Na$^+$. The dashed line is the nonlinear regression on similar experiment performed on LeuT WT (see Supplementary Fig. 2c) shown here for comparison. LeuT WT $K_D$ for leucine is $20.1 \pm 4.2$ nM (mean ± s.e.m., $n = 4$, see Supplementary Table 1). (**b**) FL-quenching values obtained by tmFRET experiments with T254V$^{tmFRET}$ (blue bars) or WT$^{tmFRET}$ (black bars) in 800 mM of the indicated ions, and 100 μM leucine (Na$^+$/Leu condition). Data points are means ± s.e.m., $n = 3$-7. **$P < 0.01$; denotes significance level from a one-way analysis of variance with *post hoc* Bonferroni test relative to the Ch$^+$ condition.

(T354V) results in spontaneous conversion of LeuT to the K$^+$-bound, likely more outward-closed conformation and that leucine, together with high levels of Na$^+$, is able to drive T354V$^{tmFRET}$ back towards its starting outward facing conformation.

To assess whether the lack of a K$^+$-induced change was due to disrupted binding or changed conformation, we performed a Schild analysis of the K$^+$-effect on [$^3$H]leucine binding to T354V$^{tmFRET}$ (Fig. 7c,d). This revealed that K$^+$ could inhibit binding of [$^3$H]leucine to T354V$^{tmFRET}$ with approximately 4-fold higher affinity than in LeuT WT suggesting that K$^+$ binding to LeuT does not require an intact Na2 site, but may be favoured by a switch of the transporter towards a more outward-closed and thereby a possibly more inward-open state. To substantiate this, we generated LeuT R5D to facilitate LeuT transition to an inward-facing state without affecting Na2 binding[32,45]. In agreement with our observation for T354V$^{tmFRET}$, Schild analysis showed a similar 4-fold higher affinity as compared with LeuT WT (Fig. 6d).

Finally, we mutated Thr354 to Asp (T354D$^{tmFRET}$) to mimic the Na2 site in the mammalian NSS proteins. In these, an aspartate is found in position 354 (Supplementary Fig. 5), otherwise the Na2 site residues are conserved. T354D$^{tmFRET}$ displayed similar tmFRET values as WT$^{tmFRET}$ in Ch$^+$ and had a K$^+$-response of similar magnitude and affinity (Supplementary Table 2 and Supplementary Fig. 5).

## Discussion
In this study, we provide, to our knowledge, the first evidence for a role of K$^+$ in regulating the function of LeuT; the most commonly used model system for NSSs. We demonstrate that K$^+$ inhibits Na$^+$-dependent binding of [$^3$H]leucine to LeuT by an apparent competitive mechanism. We also show that K$^+$ is important for LeuT function as [$^3$H]alanine uptake in LeuT proteoliposomes was markedly stimulated when internal Cs$^+$ was substituted with K$^+$. Moreover, by application of tmFRET, we show that K$^+$ binding is linked to the outward-closed/inward facing state of the transport protein. Of particular interest, K$^+$ counter-transport has previously been proposed to occur in

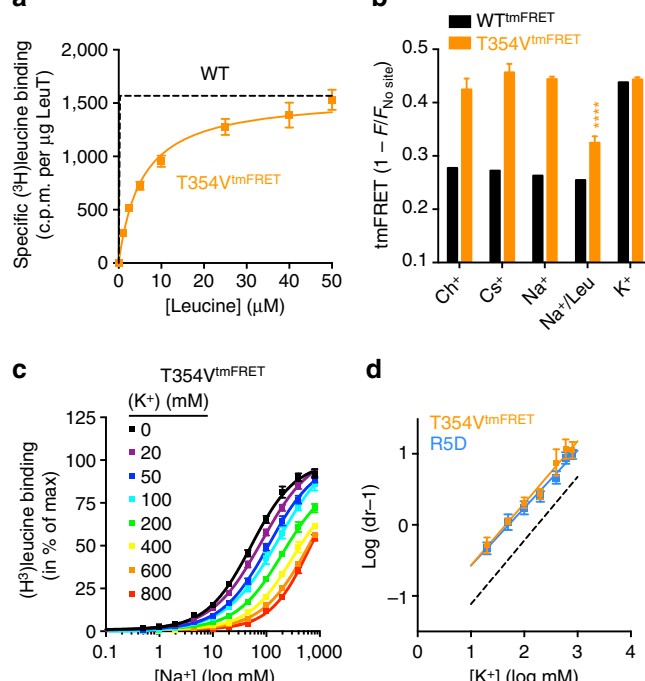

**Figure 7 | Disruption of the Na2 site potentiates K$^+$ binding.**
(**a**) Saturation binding of [$^3$H]leucine measured by the scintillation proximity assay, for the Na2-site mutant T354V$^{tmFRET}$ (orange symbols, $K_D$ $5.45 \pm 0.40\,\mu M$) in the presence of 800 mM Na$^+$. The dashed line is the nonlinear regression on similar experiment performed on LeuT WT (see Supplementary Fig. 2c) shown here for comparison. LeuT WT $K_D$ for leucine is $20.1 \pm 4.2$ nM (mean ± s.e.m., $n = 4$, see Supplementary Table 1).
(**b**) TmFRET values obtained with T354V$^{tmFRET}$ (orange bars) compared with the WT$^{tmFRET}$ background (black bars), all in 800 mM of the indicated ions, and 100 μM leucine (Na$^+$/Leu condition). T354V induces a K$^+$-like conformation even in the presence of Ch$^+$ ($0.43 \pm 0.02$), Cs$^+$ ($0.46 \pm 0.02$) or Na$^+$ ($0.44 \pm 0.004$). Only 100 μM leucine in Na$^+$ reduced the tmFRET value ($0.33 \pm 0.01$) relative to K$^+$ ($0.44 \pm 0.004$).
(**c**) Na$^+$-dependent [$^3$H]leucine binding to T354V$^{tmFRET}$ in the presence of indicated K$^+$ concentrations (substituting with Ch$^+$). (**d**) Schild plot of the EC$_{50}$ values obtained in **e** reveal the inhibition constant for K$^+$ in T354V$^{tmFRET}$ (orange squares; $K_B = 45.0$ [31.2;59.7] mM). To investigate whether the increased K$^+$ affinity could be because the T354V bias LeuT towards inward facing, we performed a similar experiment with the inward facing mutant R5D (blue squares), which possessed a similar $K_B = 47.3$ [36.7;59.6] mM). LeuT WT is shown for reference (dashed line; see Fig. 1d). Data points and tmFRET data are means ± s.e.m., $n = 3$-8. $K_B$ values are shown as mean [95% confidence interval]. ****$P < 0.0001$ based on one-way analysis of variance with *post hoc* Bonferroni's multiple test relative to the Ch$^+$ condition.

human SERT based on analyses of plasma membrane vesicles isolated from blood platelets[7,8]. However, a role for K$^+$ in the NSS transport mechanism outside of SERT has, to our knowledge, not been reported.

Visualization of atomic-scale motions is essential for elucidating the mechanistic basis for protein function. In this study, we perform FRET measurements on purified LeuT between cysteine-conjugated fluorescein and Ni$^{2+}$ bound to His-$X_3$-His motifs. The tmFRET method, which offers several advantages compared with classical FRET between two fluorophores[40], has been used before in several proteins for assessing conformational dynamics but almost exclusively in soluble proteins[36,41,42]. In LeuT, we first validated the application

of tmFRET by generating a series of tmFRET-pairs in the water-exposed EL2. Our results revealed an apparent $R_0 \sim 12$ Å, for FL and $Ni^{2+}$, which is in agreement with previous reports and supports the applicability of the method to assess shorter intramolecular distances[39,40]. Next, we constructed three tmFRET-pairs to triangulate the movement of EL4, which moves substantially between the outward- and inward-facing conformations of LeuT[17,29]. To our surprise, we observed that in the presence of $K^+$, compared with all other ions tested, tmFRET was markedly higher between EL4 and TM10, as well as between EL4 and EL2, indicating that the diameter of the extracellular permeation pathway became reduced in the presence of $K^+$. Notably, tmFRET measured in high $K^+$ (800 mM) approximately corresponded to the theoretical tmFRET value, calculated from coordinates in the outward-closed/inward-facing crystal structure (compare Fig. 2e with Supplementary Fig. 3d).

On the basis of previous EPR[44,46], cysteine accessibility[47] or smFRET[32,33,47] studies on LeuT, it has been suggested that $Na^+$ and $Li^+$ promotes an outward-open/inward-closed state of the transporter. Our tmFRET measurements, however, did not reveal any effect of $Na^+$ nor $Li^+$ (that have smaller radii than $K^+$) in comparison with the larger ions ($Rb^+$, $Cs^+$, $Ch^+$, $NMDG^+$, $Tris^+$) that do not support substrate binding. It is therefore interesting to note that $K^+$ was used as control for $Na^+$ in the previous cysteine accessibility[47] or smFRET[32,33,47] studies, indicating that the ability of $K^+$ to promote a more outward-closed conformation might have been missed. Indeed we observed that the change in tmFRET, and thus the conformational change in response to $K^+$, was attenuated by $Na^+$ and fully blocked by leucine in combination with $Na^+$. This could reflect that the $K^+$-induced, presumably outward-closed conformation is coupled to reorganization of the $Na^+$ sites as well as the substrate site. Importantly, the crystal structure of inward-facing LeuT also displayed large distortions of the substrate and $Na^+$ sites (PDB: 3TT3). We found moreover that $Na^+$ (and $Li^+$ with a lower potency) could return LeuT to the outward-open state, starting from a transporter population in an outward-closed state promoted by $K^+$. The addition of substrate potentiated the ability of $Na^+$ to return LeuT to the outward-open state.

Although the measurements did not reveal an effect of $Na^+$ on the maximum tmFRET values, $Na^+$ and $Li^+$ (as well as $K^+$ and $Rb^+$) did appear to increase $Ni^{2+}$ affinity for the EL4 His-$X_3$-His site, suggesting that these ions upon binding cause a change in the secondary structure of EL4. Indeed, $Na^+$-induced changes in EL4 have been reported with EPR spectroscopy[29], and tmFRET has proven eligible for probing changes in α-helical secondary structure[40,43]. When interpreting our data, we should also note that tmFRET between EL4 and TM10 obtained in $Na^+$ ± leucine appeared to deviate towards a more closed conformation for LeuT $WT^{tmFRET}$ compared with tmFRET values expected from crystal structures (Supplementary Fig. 3). However, tmFRET is based on ensemble measurements of a dynamic protein, which might reflect an average of conformations with a mean distance that differ from the measurements based on crystal structures. Finally, we should note that in the presence of $Na^+$, our tmFRET measurements indicated an apparent decrease in the EL2–EL4 distance on addition of leucine but an equivalent decrease was not seen between EL2 and TM10 (Figs 2e and 3a). According to the available LeuT crystal structures[11,17], a decrease could be expected in both directions, that is, in the presence of both $Na^+$ and leucine, the transporter should assume a more outward-closed configuration involving a decrease in the two distances. At this stage, we do not have an immediate explanation for this apparent discrepancy; however, crystal structures are static and may not reflect the entire ensemble of movements taking place in the protein in response to binding of ions and substrate.

To probe $K^+$ binding when LeuT is stabilized in the outward-open state, we mutated Arg30 ($R30A^{tmFRET}$) to disrupt the conserved Arg30–Asp404 salt bridge[32]. The tmFRET measurements confirmed that $R30A^{tmFRET}$ in the absence of binding ions was more open to the outside compared with $WT^{tmFRET}$. We observed no tmFRET-response to $K^+$ for $R30A^{tmFRET}$. Intriguingly, tmFRET values for $R30A^{tmFRET}$ between TM10 and EL4 were in the expected range when the Arg30–Asp404 salt bridge was disrupted (Fig. 5 and Supplementary Fig. 3). The data suggest that $K^+$ binding to the outward-facing conformation of LeuT is highly improbable under the investigated concentrations of $Na^+$ and substrate. Another possibility would be that Arg30 participates directly in $K^+$ binding; however, we find this less likely. Given our strong evidence that $K^+$ preferentially binds that outward-closed/inward-facing conformation, it is striking that when biasing the transport towards the outward-open/inward-closed conformation by mutating Arg30, the effect of $K^+$ is essentially gone (as would be expected). Moreover, as internal $K^+$ stimulates the accumulation of substrate, $K^+$ is likely to bind from the inside making it less likely that $K^+$ binds in the extracellular vestibule[17].

To probe the possible role of the Na sites in $K^+$ binding, we mutated the Na1 and Na2 site in LeuT, respectively. Thr254 was substituted with Val ($T254V^{tmFRET}$) to disrupt the Na1 site and, as expected, [³H]leucine affinity was greatly reduced in this mutant, because Na1 is required for coordination of the substrate carboxy-moiety[11]. The mean tmFRET between TM10 and EL4, however, was unchanged for $T254V^{tmFRET}$ compared with $WT^{tmFRET}$ (in $Cs^+$, $Ch^+$, $Na^+$). In 800 mM $K^+$, we observed a significantly higher, tmFRET compared with $Ch^+$ but the increase was markedly smaller than for $WT^{tmFRET}$ and so small that we could not reliably determine the $EC_{50}$ for the $K^+$ effect. Thus, it is possible that an intact Na1 site is required for $K^+$ binding but because of the functional perturbation of LeuT by the T254V mutation, it is not possible to conclude whether T254V disrupts coordination of $K^+$, or perturbs the conformational change required for $K^+$ binding. Indeed, the mutation did not ablate the $K^+$ effect completely.

To disrupt the Na2 site, we mutated Thr354 ($T354V^{tmFRET}$) and observed that tmFRET between TM10 and EL4, in its apo-form ($Ch^+$ or $Cs^+$), was identical to the $K^+$-bound conformation observed in $WT^{tmFRET}$; and approximately corresponding to tmFRET predicted from the inward-facing crystal structure (PDB: 3TT3; ref. 17). Schild plots suggested that T354V potentiated $K^+$ binding. This was also observed in the R5D mutation, where the intracellular interaction network has been destabilized[28,32]. Thus, $K^+$ binding to LeuT appears to be favoured when the Na2 site and/or inner gate are disrupted. Taken together, this suggests that Thr354 is involved in stabilizing the outward-open conformation, possibly by forming a low-barrier hydrogen-bond with the backbone carbonyl oxygen of Gly20 (compare PDB: 2A65 and 3TT1)[11,17]. We observed a similar tmFRET profile and $EC_{50}$ for $K^+$ in LeuT $T354D^{tmFRET}$; suggesting that a similar role of the Na2 site may be plausible in mammalian NSS.

Collectively, our data support a model where internal $K^+$ interacts with an NSS protein in an outward-closed/inward-facing conformation. We propose that internal $K^+$ facilitates transport by binding to LeuT in a way that inhibits rebinding of $Na^+$ and, accordingly, also substrate. This will minimize substrate efflux and result in an increased concentrative capacity of the transporter, that is, substrate can be transported against a larger chemical gradient. Unbinding of $K^+$ from LeuT will then favour

an outward-open conformation of the transporter that in turn is further stabilized by $Na^+$. Additional studies will be required to address whether $K^+$ itself is counter-transported by LeuT, as proposed for SERT[48]. Further studies are also required to elucidate whether $K^+$ interact with a broader range of NSS proteins, and thus whether $K^+$ has a general role in regulating the NSS transport cycle.

## Methods

**Implementation of tmFRET in LeuT.** To validate the method for LeuT, we constructed a series of tmFRET-pairs with defined donor–acceptor distances ($r$) in the solvent accessible α-helical stretch of EL2 (Supplementary Fig. 1a). Single cysteines were inserted in position 136, 138 or 142 (LeuT has no native cysteines), in combination with a His-$X_3$-His motif (E146H-S150H). Fluorescence spectroscopy revealed that $Ni^{2+}$ dose-dependently quenched fluorescence (F) from purified LeuT R142C$^{FL}$-E146H-S150H, and that quenching was substantially diminished in the absence of the $Ni^{2+}$ site ($F_{no\ site}$, LeuT R142C$^{FL}$, Supplementary Fig. 1b,c). To correct for dilution, collisional quenching and the inner-filter effect of $Ni^{2+}$, fluorescence intensities from the tmFRET construct (F) were normalized to the fluorescence intensities from the control construct devoid of the His-$X_3$-His site ($F_{no\ site}$). The resulting tmFRET values ($1 - F/F_{no\ site}$) for the three tmFRET pairs were plotted as a function of $Ni^{2+}$ concentration, yielding concentration-response curves with $EC_{50}$ for $Ni^{2+}$ in the micromolar range (Supplementary Fig. 1d). The tmFRET value at saturating $Ni^{2+}$ (obtained by curve fitting) closely followed a Förster distance-relationship with $R_0 \sim 12$ Å when plotted against modelled $C_\beta$–$C_\beta$ distances ($r$) between the FL transition dipole and the $Ni^{2+}$ site (Supplementary Fig. 1e). To substantiate the specificity of tmFRET, we investigated $Zn^{2+}$, a non-coloured transition metal. We observed that $Zn^{2+}$ dose-dependently increased the $EC_{50}$ for $Ni^{2+}$, and that 5 mM of $Zn^{2+}$ could block FL quenching by $Ni^{2+}$ completely (Supplementary Fig. 1f,g). In contrast, 5 mM of $Ca^{2+}$, which is not a transition metal, did not affect the $Ni^{2+}$-quenching response (Supplementary Fig. 1f,h). As expected, $Zn^{2+}$ titration did not produce tmFRET, owing to the lack of spectral overlap between $Zn^{2+}$ and FL (Supplementary Fig. 1i). Thus, $Zn^{2+}$ mimics the effect of binding to the engineered His-$X_3$-His site, without acting as a FRET acceptor.

**Generation of LeuT mutants.** cDNA encoding LeuT was cloned into a pET16b derivative vector harbouring a carboxy (C)-terminal octahistidine tag and a thrombin site. LeuT variants were generated by the QuikChange method (Agilent Technologies) using primers (Eurofins Genomics) with the following sequences: D136C (5′-CCACGTGCCCCGATTCCATTCTC-3′ and 5′-GAATCGGGGCACG TGGCGTTTG-3′); D138C (5′-CCAAACGCCACGGATCCCTGCTCCATTCTCA GACCC-3′ and 5′-GGGTCTGAGAATGGAGCAGGGATCCGTGGCGTTTGG-3′); R142C ( 5′-CGATTCCATTCTCTGCCCGTTTAAAC-3′ and 5′-GAAAGTGTTTA AACGGGCAGAGAATGG-3′); K398C (5′-GCACACCTTGTGATGTTCTTAAAC TGCTCCCTTGACGAGATGG-3′ and 5′-CCATCTCGTCAAGGGAGCAGTTTA AGAACATCACAAGGTGTGC-3′); E146H-S150H (5′-CTCAGACCCTTTAAAC ACTTTCTGTACCACTACATCG-3′ and 5′-CGATGTAGTGGTACAGAAAGTGT TTAAAGG-3′); E146H-S150H, in the R142C background (5′-GCC CCTTCAAACACTTTCTGTACCACTACATCGGAGTTCC-3′ and 5′-GGAACT CCGATGTAGTGGTACAGAAAGTGTTTGAAGGGGC-3′); K145H-Y149H (5′-CCATTCTCAGACCCTTTCATGAATTTCTGCACTCCTACATCG-3′ and 5′-GGAACTCCGATGTAGGAGTGCAGAAATTCATGAAAGGGTCTG-3′); A313H-A317H (5′-GCAAATGCGGTTCACATTGCAAAGCACGGAGCCTTTA ACC-3′ and 5′-GGTTAAAGGCTCCGTGCTTTGCAATGTGAACCGCATT TGC-3′); R5D (5′- CCATGGAAGTTAAAGACGAACACTGGGCGACGCGTCT CGG-3′ and 5′- CCGAGACGCGTCGCCCAGTGTTCGTCTTTAACTTCCA TGG-3′); R30A (5′-GGACTTGGTAATTTCCTCGCATTTCCCGTTCAAGCT GCG-3′ and 5′-CGCAGCTTGAACGGGAAATGCGAGGAAATTACCAAG TCC-3′); T254V (5′-CTGTAGGACAAATATTCTTCGTCCTGAGTCTTGGTTTT GGAGC-3′ and 5′-GCTCCAAAACCAAGACTCAGGACGAAGAATATTTGTC CTACAG-3′); T354V (5′-CCTCTTCTTCGCAGGACTCGTTTCTTCAATAGCT ATAATGCAACC-3′ and 5′-GGTTGCATTATAGCTATTGAAGAAACGAGTCC TGCGAAGAAGAGG-3′); T354D (5′-CCTCTTCTTCGCAGGACTCGATTCTT CAATAGCTATAATGC-3′ and 5′-CCTCTTCTTCGCAGGACTCGATTCTTC AATAGCTATAATGC-3′). The generated constructs were verified by DNA sequencing.

**LeuT expression and purification.** We cultivated C41-strain *Escherichia coli*, transformed with the desired LeuT constructs, in Lysogeny Broth containing 75 μg ml$^{-1}$ ampicillin at 37 °C with 180 r.p.m. shaking. At $D_{600} \sim 0.6$, expression was induced by the addition of 100 μM isopropyl β-D-1-thiogalactopyranoside with subsequent incubation at 20 °C and 180 r.p.m. shaking for 20 h. The membrane fraction was isolated by a double passage of the cells through a Basic Z Cell Disruptor (Constant Systems) at 2.30 bar and 4 °C, with subsequent ultracentrifugation at 119,000g (Beckmann Coulter) for 2 h. The membranes were resuspended in ice-cold Buffer A (50 mM Tris-HCl (pH 8.00), 30% (w/v) glycerol, 300 mM KCl, 5 mM MgCl$_2$, 1 mM tris-(2-carboxyethyl)-phosphine (TCEP)), and

LeuT was solubilized by addition of 1.0% (w/v) n-dodecyl-β-D-maltoside (DDM, Anatrace) with rotation for 1.5 h at 4 °C. LeuT was immobilized on ProBond Ni-IDA resin (Life Technologies) and incubated in Buffer B (20 mM Tris-HCl (pH 7.50), 200 mM KCl, 20% (w/v) glycerol and 0.1 mM TCEP, 0.05% (w/v) DDM) supplemented with 50 mM imidazole for 1 h at 4 °C. The resin was washed three times with ice-cold Buffer B to remove unbound impurities. Fluorescent conjugation was initiated by the slow addition of fluorescein-5-maleimide (FL, Life Technologies) to a final concentration of 200 μM. The reaction was incubated at 4 °C for 16 h, under slow rotation and in the dark. Unconjugated FL was removed by washing the LeuT-bound resin with Buffer B containing 90 mM imidazole until $A_{490}$ returned to baseline. LeuT was collected as a single peak by elution with Buffer B containing 300 mM imidazole. LeuT yields were quantified by ultraviolet–visible spectroscopy as $A_{280}$ ($\varepsilon_{LeuT} = 113.300$ M$^{-1}$ cm$^{-1}$) corrected for FL absorbance at 280 nm as per the manufacturer's instructions. The FL labelling efficiencies were quantified as $A_{490}$ ($\varepsilon_{FL} = 83.000$ M$^{-1}$ cm$^{-1}$). SDS–PAGE analysis verified labelling specificity and sample purity (see Supplementary Fig. S3). The samples were stored at $-80$ °C as single-use aliquots.

**Protein reconstitution.** Proteoliposomes were prepared from *E. coli* total lipid extract (Avanti Polar lipids Inc.) as reported[49] from detergent solubilized LeuT WT using protein:lipid (w:w) ratio of 1:100. In brief, lipid was dried under a gentle stream of nitrogen to remove the organic solvent chloroform, remaining chloroform traces were removed overnight in a rotavapor. The lipid films formed were hydrated, with internal solution containing 200 mM KCl, CsCl and NaCl and buffered with 20 mM HEPES at pH 7.5 (pH was maintained with ammonium hydroxide), with a final concentration of lipid 20 mg ml$^{-1}$. The lipid suspension was sonicated for a total of six cycles (one cycle of 15 s on/45 s off on ice), flash frozen and slowly thawed at room temperature three times. Liposomes were then extruded 11 times with a mini extruder (Avanti lipids) over a filter of pore size 400 nm. Extruded liposomes were then destabilized stepwise with 0.2% (w/v) Triton X-100 followed by purified LeuT addition. The mix was incubated at room temperature for 30 min over gentle rotation. The detergent was removed using four cycles of stepwise Bio-Beads (Bio-Rad) addition (in total 160 mg ml$^{-1}$); where first of the four cycles was performed at room temperature for 30 min with gentle rotation and remaining three were performed at 4 °C. The Bio-Beads were removed via filtration followed by proteoliposomes collection by ultracentrifugation at 120,000g for 120 min. The proteoliposome pellet was resuspended (to ~100 mg lipids per ml) in internal solution. The proteoliposomes were then stored at $-80$ °C. Before every uptake experiment, frozen liposomes were extruded as mentioned above.

**[³H]Leucine binding.** In a clear-bottom 96-well plate (Corning), 0.5 μg ml$^{-1}$ of the tested LeuT variants were mixed with 0.125 mg ml$^{-1}$ YSi-Cu His-Tag SPA beads (PerkinElmer), and [³H]leucine (PerkinElmer, 10.8 Ci mmol$^{-1}$) in Assay Buffer (20 mM Tris-HCl (pH 7.50), 0.1 mM TCEP, 0.05% (w/v) DDM) supplemented with 200 mM of the indicated cations. Sodium-site mutants (T254V and T354V) were assayed using 10 μg ml$^{-1}$ LeuT, 0.250 mg ml$^{-1}$ YSi-Cu His-Tag SPA beads, and [³H]leucine (PerkinElmer, 0.50 Ci mmol$^{-1}$) in Assay Buffer supplemented with 800 mM NaCl. Na$^+$ dependence was determined using 0.5 μg ml$^{-1}$ of the tested LeuT variants, with 0.125 mg ml$^{-1}$ YSi-Cu His-Tag SPA beads and 100 nM or 1.0 μM [³H]leucine, in Assay Buffer supplemented with varying NaCl (or LiCl). Ionic strength was maintained by substitution with KCl or ChCl as indicated. Schild plots were obtained similarly, except using 5.0 μM [³H]leucine in Assay Buffer supplemented with the indicated concentrations of KCl and NaCl. Ionic strength was maintained by substitution with ChCl. Nonspecific background was determined in the presence of 5 mM alanine or 1 mM leucine for the sodium-site mutants. Plates were sealed, mixed on an orbital shaker for 30 min at room temperature, and incubated for ~16 h at 4 °C. Counts per minute (c.p.m.) were recorded on a 2450 MicroBeta² microplate counter (PerkinElmer), and no further increase in c.p.m. was observed for longer periods of incubation.

**[³H]Alanine uptake.** [³H]Alanine (PerkinElmer) uptake was carried out at 30 °C as reported previously[50]. Proteoliposomes were prepared as described above. The internal solution of proteoliposomes contained either 200 mM KCl or 200 mM CsCl, 20 mM HEPES at pH 7.5. Background binding was determined using gradient-free proteoliposomes reconstituted in 200 mM NaCl. The uptake was initialized by diluting the proteoliposomes into external buffer (200 mM NaCl, 20 mM HEPES at pH 7.50, 0.5 μM [³H]alanine), at 30 °C. The reaction was stopped at indicated time by 10-fold dilution into ice cold stopping buffer (200 mM KCl, 20 mM HEPES at pH 7.50). Proteoliposomes were then collected over nitrocellulose filters of 0.22 μm pore size. The filters were washed three times with 4 ml of ice-cold stopping buffer. The dried filters were immersed into 3 ml of scintillation cocktail. Radioactivity was measured in a 2450 MicroBeta² microplate counter after mixing on a shaker for 2 h. Total amount of incorporated functional LeuT was assessed for each reconstitution by solubilization of proteoliposomes in DDM (1%) and measurement of total [³H]leucine binding.

**[³H]Alanine efflux.** The proteoliposomes were prepared as described above with internal solution of either 200 mM KCl or 200 mM CsCl (in 20 mM HEPES,

pH 7.50). Proteoliposomes were preloaded for 10 min in external buffer (200 mM NaCl, 20 mM HEPES, pH 7.50) with 50 nM [³H]alanine (79.5 Ci mmol$^{-1}$). At $t = 0$, half of the reactions were stopped to determine maximal [³H]alanine uptake. The remaining proteoliposomes were diluted five times in external buffer supplemented with 500 nM unlabelled alanine and incubated for 30 min to allow efflux. Uptake during the 30 min incubation was assessed in parallel by 10 min preloading in 50 nM unlabelled alanine, followed by 5 × dilution in external buffer containing 500 nM [³H]alanine (7.95 Ci mmol$^{-1}$) and incubated for 30 min. All the efflux experiments were performed at 30 °C. The reactions were stopped by filtration on nitrocellulose filters and washed with ice-cold stop buffer (200 mM KCl, 20 mM HEPES, pH 7.50). The filters were immersed into 3 ml scintillation cocktail solution, incubated for 24 h on shaker and recorded on a 2450 MicroBeta² microplate counter.

**Fluorescence spectroscopy.** FL labelled LeuT variants generated for tmFRET experiments were diluted to 0.5 μg ml$^{-1}$ (8.4 nM) in Fluorescence Buffer (20 mM Tris-HCl (pH 7.50), 0.1 mM tris-(2-carboxyethyl)-phosphine (TCEP) and 0.05 DDM (Anatrace)) that was supplemented with NaCl, KCl, LiCl, RbCl, CsCl, NMDG or Tris/MES as indicated. Fluorescence recordings were carried out at 23 °C on a FluoroMax-2 or a FluoroMax-4 Spectrofluorometer (Horiba Scientific). Emission spectra were recorded as blank corrected emission intensities using excitation at 492 nm with 3 nm excitation and emission band-pass. Constant wavelength recordings were obtained as the peak emission intensity at 520 nm with excitation at 492 nm and with 5 nm excitation and emission band-pass. The tmFRET Ni²⁺-titration experiments were performed by recording the peak fluorescence intensity in response to successive additions of small aliquots of NiCl₂. The tmFRET ion titration experiments were performed in Fluorescence Buffer supplemented with 750 μM NiCl₂ (or ZnCl₂ for control) and 0–750 mM of NaCl, LiCl or KCl as specified. The samples were incubated for 30 min at room temperature before sample fluorescence intensity was recorded. Competing effects of ions and substrates on the K⁺-response was investigated by performing the KCl titration experiment in buffers containing 100 mM of NaCl, LiCl or CsCl and in the presence or absence of 50 μM leucine. The tmFRET NaCl and LiCl back-titration experiments were performed similarly, except that KCl was kept at constant concentration of 600 mM to induce the K⁺ bound conformation. In all the tmFRET ion titration experiments, ionic strength was maintained by substituting the titrated cation with ChCl. Background fluorescence from buffer and ligands was not detectable at 520 nm. Furthermore, thrombin cleavage of the C-terminal his-tag that binds Ni²⁺ did not change tmFRET recordings for the generated tmFRET variants.

**Absorbance spectra.** Ultraviolet–visible absorbance spectra of Ni²⁺, Zn²⁺ and Ca²⁺ diluted in Fluorescence Buffer with 200 mM NaCl, were obtained on a NanoQuant Infinite M200 (Tecan). The obtained absorbance values (A) were converted to molar extinction coefficients ($\varepsilon$) by the Lamber–Beer law: $\varepsilon = A$ (c d)$^{-1}$, where c is the molar concentration and d is the light path length in cm.

**Data analysis.** All Na⁺-dependence data were fitted to the Hill equation to derive potency estimates (EC₅₀ values), that is, the concentration of Na⁺ (A) that gives the half-maximal binding of [³H]leucine, defined by the lower and upper (B$_{max}$) asymptotes of the binding curves. We constructed Schild plots from EC₅₀ values for Na⁺ and Li⁺ in the presence (A′) and absence (A) of the 'antagonist' K⁺ (B) giving the dose ratios (dr = A′/A). Thus, values of log(dr − 1) were plotted as a function of the corresponding log[B] values. K⁺ affinity ($pK_B$) values were calculated with the Schild equation:

$$\log(\mathrm{dr} - 1) = \log[B] - K_B \tag{1}$$

All the experiments were repeated at least three times unless otherwise stated. Data points are given as means ± s.e.m. or means with 95% confidence intervals. Kinetic and equilibrium constants and tmFRET values were obtained using non-linear regression algorithms in Prism 5.0 or 6.0 (GraphPad Software). Statistical analyses were performed using Student's t-test or one-way analysis of variance multiple comparison test as appropriate.

**Analysis of fluorescence data.** Fluorescence measurements were corrected for dilution, the inner-filter effect of Ni²⁺ and collisional quenching as described[42] by normalizing fluorescence intensities from the tmFRET construct ($F$) to fluorescence intensities from the control construct devoid of the His-$X_3$-His site ($F_{no\ site}$). The corrected FL quenching ($1 - F/F_{no\ site}$) plotted as a function of log [Ni²⁺] and fitted to a single-site model yielded the apparent tmFRET efficiency ($E$). Theoretical values of $E$ were calculated using the Förster equation: $E = 1 + (R/R_0)^6$, where $R_0 = 12$ Å (ref. 40) and donor–acceptor distances ($R$) were determined from published X-ray crystal structures of LeuT (PDB code: 2A65; 3TT1; 3TT3)[11,17].

**Data availability.** The data that support the findings of this study are available from the corresponding author upon reasonable request.

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

## Acknowledgements

The pET16b LeuT WT plasmid was kindly provided by Dr Eric Gouaux, Vollum Institute, Oregon Health Science University. Lone Rosenquist is thanked for excellent technical assistance. Professor Jonathan Javitch, Professor Harel Weinstein, Professor Baruch Kanner, Associate Professor Søren G.F. Rasmussen and Dr Lars Borre are thanked for discussions and helpful suggestions. The work was supported in part by the Danish Independent Research Council – Sapere Aude (0602-02100B) (C.J.L.), the Lundbeck Foundation (R108-A10755) (C.J.L.), the National Institute of Health Grants P01 DA 12408 (U.G.), the Lundbeck Foundation Center for Biomembranes in Nanomedicine (U.G.), the UNIK Center for Synthetic Biology (U.G. and C.J.L.), the European Community's Seventh Framework Programme FP7/2007–2013 HEALTH-F4-2007-201924 (U.G.) and bioSYNergy, University of Copenhagen's Excellence Program for Interdisciplinary Research (C.J.L. and U.G.), the Austrian Science Fund/FWF (F3506) (H.H.S.), Graduate School of Health and Medical Science, University of Copenhagen (C.B.B.), The Bikuben Foundation New York (C.B.B.) and the Intramural Research Program of the NIH, NIDA (ZIA DA000606-01) (L.S.).

## Author contributions

C.B.B., U.G. and C.J.L. designed the study. C.B.B. generated and purified LeuT variants with support from J.S.M. C.B.B. established protocols for tmFRET and radiotracer binding experiments and performed the majority of experiments with support from J.S.M. A.S., J.S.M. and S.G.S. performed reconstitution and uptake experiments with collated binding controls. C.B.B., U.G. and C.J.L. designed the experiments and interpreted the data with support from L.S., J.S.M., H.H.S. and A.S. C.B.B., U.G. and C.J.L. wrote the manuscript and all the authors commented.

## Additional information

**Competing financial interests:** The authors declare no competing financial interests.

