## [Peer Review File · Nature Communications]

Reviewer #1 (Remarks to the Author)

LeuT is a prokaryotic protein that is structurally similar to the eukaryotic neurotransmitter:sodium symporters and has been crystalized in multiple conformational states including outward facing and inward facing conformations. This paper shows that LeuT is regulated by K^+ , inhibiting the binding of Na^+ and Leu and increasing the transport of Ala. Using a powerful method called transition metal ion FRET (tmFRET), these authors uncover details of the structural rearrangements produced by K^+ . They show that, unlike other monovalent cations, K^+ causes a conformational change in extracellular loop 4 (EL4) that acts like a lid sealing the extracellular gate. The collected results suggest a mechanism similar to that previously proposed for the serotonin transporter SERT where K^+ binding in the inward facing conformation prevents rebinding of neurotransmitter and promotes return of the transporter to the outward facing conformation for binding of Na^+ and neurotransmitter.

The results are convincing and interesting. The tmFRET experiments are particularly elegant and provide mechanistic details that were not seen in previous double electron-electron resonance (DEER), single molecule FRET (smFRET) and site-directed fluorescence quenching spectroscopy experiments. The paper, in general, is well written, but the large number of results and their mechanistic implications are sometimes hard to follow. Below are some suggestions to help improve the presentation.

Suggestions:

My biggest suggestion is to put the cyclic model summarized in the last paragraph of the paper, earlier in the paper, either in the introduction or at the beginning of the discussion. An associated figure would also be useful. It would help a lot to be able to understand the results in the context of the model.

It is stated a number of times that " K^+ competitively inhibits Na^+ -dependent binding of substrate". While it is clear that K^+ inhibits Na^+ and Leu binding, it is not clear to me that the mechanism is "competitive" which to me means sharing a common site. In fact, it is shown that K^+ produces a conformational change that is likely to exhibit a lower affinity for Na^+ and Leu. This sounds like a noncompetitive mechanism. Also, it seems possible that K^+ is competing for one of the Na^+ binding sites, but is that technically competitive binding with Leu? The use of the term "competitive" should be explained or removed.

A related comment...the Schild analysis used several times in the paper should be explained (and a reference given). Does Schild analysis discriminate between competitive and noncompetitive inhibition? Perhaps if I understood it better, I would not have the comment above.

It should be pointed out in the results that the binding and tmFRET experiments are all done on transporter in detergent. Is there evidence that the conformational change in detergent is structurally and energetically similar to the transition in membranes?

Page 5. "...substituted with K+..." It was not clear what this meant at first.

Page 6 "tmFRET provides an ideal sensitivity range (~6-24 Å)..." This range seems overly broad and would require detecting FRET efficiencies of only about 1.5%. I suggest 8-18 Å.

Page 7. "the apparent affinities for the Ni²⁺ response were significantly increased ...(reflected in increased pEC₅₀ values, Fig. 2f)." I think the first "increased" should be "decreased".

Page 8. "K⁺ produced a ~22% increase in tmFRET with an EC₅₀ for K⁺ (Fig. 4a and Table 1)." What was the EC₅₀ for K⁺?

Page 9. A 25-fold increase..." Decrease?

Page 10. Heading: "LeuT trapped in the outward-open conformation blocks K⁺ binding." Do the data support no binding or just no effect of binding?

Fig. 6a and 7a. The dotted line labeled "WT" should be explained, and the affinity for WT should be stated in the text.

Reviewer #2 (Remarks to the Author)

This paper reports evidence for a previously unreported interaction between the amino acid transporter LeuT and K⁺ ions. LeuT is of general interest because it has been a structural and, to some extent, a functional model for the mammalian Na⁺-dependent neurotransmitter transporters that transport dopamine and serotonin. The results reported here clearly establish that K⁺ ions (at very high concentrations) can subtly (3-fold) interfere with the protein's ability to bind its normal substrate. The authors further show that K⁺ competitively influence the protein's interaction with Na⁺, the ion which drives transport. However, K⁺ can not itself support substrate binding. Surprisingly in this context, adding K⁺ seems to enhance the transport activity of LeuT. Further experiments explore the structural basis of the K⁺ effect. Assays using transition metal FRET (tmFRET) support the authors' argument for a K binding site, with similar interactions as seen with the binding assays. Further experiments with mutant LeuT's with alterations in the S2 site or Na binding sites are more difficult to interpret in terms of conformational changes and the effects of K⁺. In general, the manuscript is well written and clear, and the experiments seem carefully performed, with data of high quality. The evidence for a low affinity K binding site is strong; however there is more difficulty interpreting the role of this site in the transport process and understanding its influence on the conformation of LeuT. In general, there are problems developing a consistent interpretation of the data here.

Major issues:

1. TM FRET. I'm not sure how to interpret the results of the two different experimental TM FRET pairs shown in Figures 2 and 3a. The data in Figure 2, with FRET labels on the TM90 loop and EL4 show a FRET change only in the presence of K⁺ (with or without Leu). In contrast, Figure 3a, with labels in EL2 and EL4 show changes either in Na/Leu, or K (with or without Leu). The authors point out this change in the text, but they do not seem to incorporate this difference in their interpretation (Page 8, 2nd full paragraph, discussion), dwelling on the consistent effect of K instead. If Na is indeed promoting an outward facing state, how can these results be explained? Shouldn't EL2 and EL4 move further apart in this situation? Shouldn't Na and K promote opposite changes in FRET?
2. R30A mutation. The authors use the R30A mutation to "probe K⁺ binding when LeuT is locked in the outward-open state" but it is not at all clear that the protein is indeed "locked". Indeed, the data in the single-molecule paper referred to show a distribution that suggests occupancy of both inward and outward facing states. This residue also contributes to the lining of the "S2" site in leuT, a secondary ligand binding site. Could the mutation itself be preventing K binding, rather than isolating a state that disallows K binding? I don't think the data provide a conclusive interpretation here.
3. Finally, the model proposed in the last paragraph of the paper is inconsistent with the data presented. The data presented in Figure 1e clearly demonstrate that K⁺ is not directly coupled to transport, as suggested by the proposal that K⁺ itself is counter-transported by LeuT, since transport occurs efficiently in the absence of K (better than when it is present!, Figure 1e). Also, given all the evidence that K seems to stabilize a particular state, it seems difficult to imagine how the same binding process would instead speed a transition during transport. However, if the authors could provide more direct evidence for their model it might be more convincing. The model generates easily testable predictions that would address the role of K in transport-for example, external K should promote efflux of Na and substrate from preloaded vesicles.

Reviewer #3 (Remarks to the Author)

In this important manuscript the Authors provide strong evidence that LeuT, which represents paradigm for the neurotransmitter:sodium:symporters, can interact with potassium and that this interaction favors the inward-facing conformation of LeuT. The motivation for this study is an almost 40 year old observation that the serotonin transporter SERT could mediate potassium-dependent countertransport of the neurotransmitter and that protons could replace the potassium. Using transition metal ion FRET as a main tool, the Authors convincingly show that potassium ions can compete with sodium ions and favor the inward-facing conformation of LeuT. Moreover they observe a stimulation of transport in proteoliposomes by internal potassium. I have only one major comment. Because there was uptake of [3H]-alanine by even in the absence of internal potassium-just like in SERT- it is quite likely that, also in LeuT, protons can functionally replace potassium. It is therefore very important to perform several of the experiments in the absence of potassium, but at lowered pH. This should not only include some of the transition metal ion FRET measurements, but also some uptake experiments in reconstituted proteoliposomes to see if stimulation of transport by an outward directed proton gradient (acidic inside) can be observed in the absence of potassium.

Minor issues:

P.3 line8: replace "several" by "all".

P8 last line from below: please enter the value for the EC50.

Fig. 1 Legend, line 12: "proteoliposomes containing..."/

Fig. 4 Legend, line 9 "purple" looks "pink" to me.

Reviewer #1

The results are convincing and interesting. The tmFRET experiments are particularly elegant and provide mechanistic details that were not seen in previous double electron-electron resonance (DEER), single molecule FRET (smFRET) and site-directed fluorescence quenching spectroscopy experiments. The paper, in general, is well written, but the large number of results and their mechanistic implications are sometimes hard to follow. Below are some suggestions to help improve the presentation.

We thank the reviewer for this positive comment about our work

My biggest suggestion is to put the cyclic model summarized in the last paragraph of the paper, earlier in the paper, either in the introduction or at the beginning of the discussion. An associated figure would also be useful. It would help a lot to be able to understand the results in the context of the model.

We thank the reviewer for reminding us of this important issue. Introducing the model earlier in the paper will indeed provide the reader with a better perspective of the results that could make the sequence of experiments easier to follow.

Accordingly, to address the reviewer's comment, we have revised the last paragraph of the introduction (page 4):

“Our data suggest altogether that K^+ could play a canonical role in regulating the function of NSS proteins. Specifically, we propose a model in which internal K^+ facilitates substrate turnover by inhibiting substrate rebinding in the outward-closed/inward-facing conformation. This will inhibit substrate efflux and result in an increased concentrative capacity of the transporter and, thus, substrate transport against a larger chemical gradient.”

However, at the present stage we would prefer not to include an associated figure. Despite the fact that our results reveal a clear role of K^+ in the transport process, we are concerned that a drawn model could be perceived as being too bold with little room for further discussion.

It is stated a number of times that " K^+ competitively inhibits Na^+ -dependent binding of substrate". While it is clear that K^+ inhibits Na^+ and Leu binding, it is not clear to me that the mechanism is "competitive" which to me means sharing a common site. In fact, it is shown that K^+ produces a conformational change that is likely to exhibit a lower affinity for Na^+ and Leu. This sounds like a noncompetitive mechanism. Also, it seems possible that K^+ is competing for one of the Na^+ binding sites, but is that technically competitive binding with Leu? The use of the term "competitive" should be explained or removed. A related comment...the Schild analysis used several times in the paper should be explained (and a reference given). Does Schild analysis discriminate between competitive and noncompetitive inhibition? Perhaps if I understood it better, I would not have the comment above.

We understand the confusion in this matter but would still argue that our data support a competitive mechanism. If the mechanism is competitive, a lower affinity for Na^+ by the addition of K^+ is exactly what would be expected. In contrast, if the mechanism is non-competitive, affinity will be preserved but B_{max} will decrease. Importantly, according to current literature, Schild analysis is the best tool for analyzing competitive antagonism (see reviews by Wyllie & Chen (2007) Br. J. Pharmacol. or Lew & Angus (1995)

TiPS). As shown in Fig. 1c, addition of K^+ causes right-shifts in the Na^+ -dependence curves for [3H]leucine binding. This can be visualized in a Schild plot (Fig. 1d). If the mechanism is competitive, the Schild plot will be linear (as seen in Fig. 1d). However, we agree with the reviewer that, if K^+ is competing for one of the Na^+ binding sites, this may not truly reflect competitive binding with leucine. Thus, it is more correct to say that our data only directly support competition between Na^+ and K^+ . To avoid misunderstandings, we have therefore in the revised manuscript restricted our use of the word ‘competitive’ and now only use it when we refer to K^+ against Na^+ . To better explain the Schild plot, we have added the following to the methods section (p23):

“All Na^+ -dependence data were fitted to the Hill-equation in order to derive potency estimates (EC_{50} values), that is, the concentration of Na^+ (A) that gives the half-maximal binding of [3H]leucine, defined by the lower and upper (B_{max}) asymptotes of the binding curves. We constructed Schild plots from EC_{50} values for Na^+ and Li^+ in the presence (A') and absence (A) of the “antagonist” K^+ (B) giving the dose ratios ($dr = A'/A$). Thus, values of $\log(dr - 1)$ were plotted as a function of the corresponding $\log[B]$ values. K^+ affinity (pK_B) values were calculated with the Schild equation: $\log(dr - 1) = \log[B] - K_B$ ”

Moreover, the following description of the Schild Plot has been added to Results (p5):

“...and performed a Schild analysis of these data showing the shifts in EC_{50} as a function of [K^+]. The Schild analysis provides information whether or not antagonism is competitive in nature: if the regression in a Schild plot is linear with a slope of 1, then the antagonism is competitive³⁸. When competitive antagonism is observed, it also allows for determination of the affinity (K_B) of the antagonist. Here, the Schild plot revealed a linear correlation with a slope of 0.9 [0.8;1.0] (mean [95% CI]) and an equilibrium constant (K_B) for K^+ of 176 [153;203] mM (**Fig. 1d**). This suggests that K^+ competitively inhibits Na^+ -binding to LeuT WT.”

It should be pointed out in the results that the binding and tmFRET experiments are all done on transporter in detergent. Is there evidence that the conformational change in detergent is structurally and energetically similar to the transition in membranes?

We agree with reviewer’s point that this is not explicitly stated. We have inserted the following sentence in the beginning of Results (page 5):

“...using the scintillation proximity assay (SPA) on detergent solubilized protein as previously reported³⁷. Note that all subsequent experiments are performed on LeuT in detergent (dodecyl- β -D-maltoside, DDM) unless otherwise stated.”

There are several studies supporting that in LeuT similar conformational transitions might take place in detergent micelles as in lipid membranes. EPR/DEER studies of LeuT in proteoliposomes (Claxton et al. (2010) NSMB) and in the detergent solubilized state (Kazmier et al. (2014) NSMB) reported similar distance distributions and dynamics between EL4 and EL6. Additionally, these studies reported distances distributions at the extracellular face that were in close agreement with X-ray crystal structures of LeuT. Furthermore, X-ray crystal structures of LeuT obtained in either bicelles or detergent were almost identical (Wang et al. (2012) NSMB). We show that distances calculated from tmFRET measurements, performed on detergent solubilized LeuT, are also in close agreement with modeled distances derived from crystal structures of LeuT (see SI Fig. 3). Thus, we anticipate that the structural changes, reported by tmFRET for

detergent solubilized LeuT, should to a large extent reflect the structural changes encountered in the context of a membrane.

Page 5. "...substituted with K+..." It was not clear what this meant at first.

We thank the reviewer for pointing this out. The explanation of the experiment was also inadequate. We have changed it to the following (page 5):

"To assess the effect of K⁺ on LeuT, we performed Na⁺-dependent [³H]leucine binding where Na⁺ was substituted with either K⁺ or choline (Ch⁺) to maintain a total ionic concentration of 200 mM."

Page 6 "tmFRET provides an ideal sensitivity range (~6-24 Å)..." This range seems overly broad and would require detecting FRET efficiencies of only about 1.5%. I suggest 8-18 Å.

We agree and have changed the range accordingly.

Page 7. "the apparent affinities for the Ni²⁺ response were significantly increased ...(reflected in increased pEC₅₀ values, Fig. 2f)." I think the first "increased" should be "decreased".

We have double-checked this. We believe 'increased' is the right term here. Relative to the non-binding ions, the Ni²⁺ affinity is increased (in ChCl = 79 μM; in NaCl = 29 μM). The EC₅₀ value is decreased, but here it is stated as pEC₅₀ which is then increased.

Page 8. "K⁺ produced a ~22% increase in tmFRET with an EC₅₀ for K⁺ (Fig. 4a and Table 1)." What was the EC₅₀ for K⁺?

Thank you for pointing out this typo. Then sentence is now complete (page 9):

"K⁺ produced ~22% increase in tmFRET with an EC₅₀ for K⁺ of ~260 mM (Fig. 4a and Table 1)"

Page 9. A 25-fold increase..." Decrease?

Definitely 'decrease' here ☺. We have changed accordingly. Thanks.

Page 10. Heading: "LeuT trapped in the outward-open conformation blocks K⁺ binding." Do the data support no binding or just no effect of binding?

We propose that it is binding as well as effect. According to the literature (see e.g. reviews by Wyllie & Chen (2007) BJP, Lew & Angus (1995) TiPS or Colquhoun (2007) TiPS) a Schild plot assesses K⁺ binding, which for the R30A mutant yields a K_B for K⁺ of more than 1 M!

Fig. 6a and 7a. The dotted line labeled "WT" should be explained, and the affinity for WT should be stated in the text.

This piece of information was missing in the figure legends. We have now added the following to legends of both Fig. 6 and Fig. 7:

“The dashed line is the non-linear regression on similar experiment performed on LeuT WT (see **Supplementary Fig. 2c**) shown here for comparison. LeuT WT K_D for leucine is 20.1 ± 4.2 nM (mean \pm s.e.m., $n = 4$, see **Supplementary Table 1**).”

Reviewer #2

In general, the manuscript is well written and clear, and the experiments seem carefully performed, with data of high quality. The evidence for a low affinity K binding site is strong; however there is more difficulty interpreting the role of this site in the transport process and understanding its influence on the conformation of LeuT. In general, there are problems developing a consistent interpretation of the data here.

We appreciate the positive and insightful comments by the Reviewer. We agree that the K^+ affinity might appear low but still the affinity is in a range where it is likely to play a physiological role in the transport process. As outlined, below we believe that our revisions should aid interpreting of our data.

Major issues:

1. TM FRET. I'm not sure how to interpret the results of the two different experimental TM FRET pairs shown in Figures 2 and 3a. The data in Figure 2, with FRET labels on the TM90 loop and EL4 show a FRET change only in the presence of K^+ (with or without Leu). In contrast, Figure 3a, with labels in EL2 and EL4 show changes either in Na/Leu, or K (with or without Leu). The authors point out this change in the text, but they do not seem to incorporate this difference in their interpretation (Page 8, 2nd full paragraph, discussion), dwelling on the consistent effect of K instead. If Na is indeed promoting an outward facing state, how can these results be explained? Shouldn't EL2 and EL4 move further apart in this situation? Shouldn't Na and K promote opposite changes in FRET?

Our results indicate that LeuT in the absence of binding cations, such as Na^+ and K^+ , assumes an outward-open conformation. Furthermore, the unbound conformation was according our tmFRET data for both the EL4 to TM10 and EL4 to EL2 measurements indiscernible from the Na^+ -bound conformation. While it is correct that previous studies of LeuT conformational dynamics have indicated that Na^+ is required for transition to the outward-open conformation (see e.g. Claxton et al., (2010) NSMB; Kazmier et al., (2014) NSMB; Zao et al., (2010+2011) Nature; Tavoulari et al., (2016) JBC), it is important to emphasize that these studies used K^+ as reference cation, under the assumption that K^+ does not interact with LeuT. This makes it complicated to compare the data with the ones presented here. Importantly, we show that Na^+ competes with the ability of K^+ to induce the inward-facing conformation (see Fig. 4) and, thus, our observation does agree with the reputed role of Na^+ in stabilizing the outward-open conformation of LeuT. However, we assert that under the experimental conditions tested in our study, which are also comparable to previous biophysical studies of LeuT, Na^+ is not *per se required* to induce the outward-open conformation in LeuT, but its binding will stabilize the conformation. The issue is discussed on page 15, 1st paragraph (no changes from original manuscript).

As stressed by the Reviewer, we observed in the presence of Na^+ an apparent decrease in the EL2-EL4 distance upon addition of leucine but an equivalent decrease was not seen between EL2 and TM10. Even

though it is entirely possible to have movements in the EL2-EL4 direction without a *distance change* in the EL4-TM10 direction, a decrease would be expected in both directions in the presence of both Na⁺ and leucine according to the available LeuT crystal structures (Yamashita et al. (2005) Nature; Krishnamurthy et al. (2012) Nature), i.e. the transporter should assume a more outward-closed configuration involving a decrease in the two distances in the presence of both Na⁺ and leucine. Importantly, we would also expect the apo-form of LeuT to be inward facing (Krishnamurthy et al. (2012) Nature). At this stage, we do not have an immediate explanation for this apparent discrepancy; nonetheless, it should be noted that the action of Na⁺ on LeuT indeed is more complex than hitherto believed (as discussed in the paragraph above). In addition, the current crystal structures are static, under the influence of Fab-fragments and mutations (in Na-sites), and may not reflect the entire ensemble of movements taking place in the protein in response to ligand binding. Overall, we still find it fair, given the focus of our paper, to emphasize the consistent effect of K⁺ on the transport protein.

To address the issue we added the following to the Discussion (page 16):

“Finally, we should note that in the presence of Na⁺, our tmFRET measurements indicated an apparent decrease in the EL2-EL4 distance upon addition of leucine but an equivalent decrease was not seen between EL2 and TM10 (**Fig. 2e and 3a**). According to the available LeuT crystal structures^{12,18}, a decrease could be expected in both directions, i.e. in the presence of both Na⁺ and leucine, the transporter should assume a more outward-closed configuration involving a decrease in the two distances. At this stage, we do not have an immediate explanation for this apparent discrepancy; however, crystal structures are static and may not reflect the entire ensemble of movements taking place in the protein in response to binding of ions and substrate.”

2. R30A mutation. The authors use the R30A mutation to "probe K⁺ binding when LeuT is locked in the outward-open state" but it is not at all clear that the protein is indeed "locked". Indeed, the data in the single-molecule paper referred to show a distribution that suggests occupancy of both inward and outward facing states. This residue also contributes to the lining of the "S2" site in leuT, a secondary ligand binding site. Could the mutation itself be preventing K binding, rather than isolating a state that disallows K binding? I don't think the data provide a conclusive interpretation here.

We agree with the reviewer's point that the R30A mutation does not 'lock' LeuT in the outward-open state. Rather it stabilizes, or bias, LeuT towards an outward-open state. We have changed the wording throughout the text to a more modest statement using words as 'bias towards' and 'stabilize' instead of 'locking'

The possibility of Arg30 participate in K⁺ binding is an interesting idea. Indeed, solely based on our experimental observations we cannot completely rule out that this is not a valid possibility. However, we believe that the possibility is less likely. First, we have evidence supporting conformationally selective binding of K⁺ to the transporter in the outward-closed/inward-facing conformation. The most important evidence for this includes: i) K⁺-induced closure of the transporter to the outside according to the tmFRET measurements; ii) increased K⁺-potency in mutations favoring the outward-closed/inward-open conformation (T354V and R5A). In this context, it is striking that when biasing LeuT towards the outward-open/inward-facing conformation (R30A) the effect of K⁺ is essentially gone (as would be expected). Secondly, internal

K⁺ stimulates the accumulation of substrate, suggesting that K⁺ binds from the inside making it less likely to access the extracellular vestibule.

To address the reviewer's comment we have inserted the following the discussion (page 16):

“Another possibility would be, that Arg30 participates directly in K⁺-binding; however, we find this less likely. Given our strong evidence that K⁺ preferentially binds that outward-closed/inward-facing conformation, it is striking that when biasing the transport towards the outward-open/inward-closed conformation by mutating Arg30, the effect of K⁺ is essentially gone (as would be expected). Moreover, since internal K⁺ stimulates the accumulation of substrate, K⁺ is likely to bind from the inside, making it less likely that K⁺ binds in the extracellular vestibule.”

3. Finally, the model proposed in the last paragraph of the paper is inconsistent with the data presented. The data presented in Figure 1e clearly demonstrate that K⁺ is not directly coupled to transport, as suggested by the proposal that K⁺ itself is counter-transported by LeuT, since transport occurs efficiently in the absence of K (better than when it is present!, Figure 1e). Also, given all the evidence that K seems to stabilize a particular state, it seems difficult to imagine how the same binding process would instead speed a transition during transport. However, if the authors could provide more direct evidence for their model it might be more convincing. The model generates easily testable predictions that would address the role of K in transport-for example, external K should promote efflux of Na and substrate from preloaded vesicles.

We appreciate these insightful comments however, we do not follow the Reviewer's interpretation of the data in Fig. 1e. We agree that transport can occur in the absence of K⁺ (suggesting that it may not be directly coupled) but indeed transport is larger when K⁺ is present inside the proteoliposomes? Moreover, we do not find it difficult to imagine that K⁺ can stimulate transport by promoting an outward-closed/inward-facing conformation. Likely, our use of the word 'stabilize' is misleading and too strong. In essence, our data suggest that K⁺ preferentially interacts with/promotes this configuration and in doing so, it exerts its action on the transport process (see below). In the revised version of the manuscript we avoid using 'stabilize' and instead say either 'preferentially binds/interacts' or 'promotes' an outward-closed/inward-facing conformation.

To specifically address the question raised by the Reviewer, we have performed an additional experiment that we believe provide further support for our suggestive model. The results of this experiment are now inserted into a revised Fig. 1 (Fig. 1f). In short, we preloaded proteoliposomes containing either 200 mM K⁺ or Cs⁺ with [³H]alanine followed by dilution in a buffer with 10x the concentration of unlabeled alanine. We then measured the amount of [³H]alanine efflux relative to the loaded amount. The data show that internal K⁺ significantly decreases efflux of [³H]alanine under these conditions over a 30 min period. We believe that these results support the idea that internal K⁺ facilitates transport by binding to LeuT in a way that inhibits rebinding of Na⁺ and, accordingly, also substrate. This should reduce substrate efflux and result in an increased concentrative capacity of the transporter, i.e. substrate can be transported against a larger chemical gradient. Unbinding of K⁺ from LeuT will then favor an outward-open conformation of the transporter that in turn is further stabilized by Na⁺.

To accommodate the observation, we have added the following to the manuscript:

Results (page 6):

“Another possible interpretation of the result would be that internal K^+ facilitates transport by binding to LeuT in a way that inhibits rebinding of Na^+ and, accordingly, also substrate. This should reduce substrate efflux and result in an increased concentrative capacity of the transporter, i.e. substrate can be transported against a larger chemical gradient. To address this question, we performed an efflux experiment. Proteoliposomes were preincubated with [3H]alanine (50 nM) for 10 min and efflux measurements were initiated by diluting the proteoliposomes five times into a Na^+ buffer with a high concentration (500 nM) of unlabeled alanine to prevent any reuptake of released [3H]alanine. The amount of [3H]alanine efflux was assessed by comparing the remaining internal [3H]alanine after 30 min to the initial amount. In agreement with this hypothesis, [3H]alanine efflux was significantly lower for K^+ -containing proteoliposomes compared to control (Fig. 1f)”

Legend to Fig 1:

“(f) Efflux experiments showing the relative amount of [3H]alanine released from proteoliposomes over 30 min with K^+ or Cs^+ (200 mM) on the inside causing an efflux of $35 \pm 2 \%$ and $53 \pm 6 \%$, respectively (means \pm s.e.m., n=6). Data are given as % of initial amount of [3H]alanine in the proteoliposomes. Proteoliposomes were preincubated with [3H]alanine (50 nM) for 10 min and efflux measurements were initiated by a 5x dilution into a Na^+ buffer containing a high concentration (500 nM) of unlabeled alanine to prevent any reuptake of released [3H]alanine. ****p < 0.01, unpaired t-test**”.

Methods (page 22):

“**[3H]Alanine efflux.** The proteoliposomes were prepared as described above with internal solution of either 200 mM KCl or 200 mM CsCl (in 20 mM HEPES, pH 7.50). Proteoliposomes were preloaded for 10 min in external buffer (200 mM NaCl, 20 mM HEPES, pH 7.50) with 50 nM [3H]alanine ($79.5 \text{ Ci mmol}^{-1}$). At t=0, half of the reactions were stopped to determine maximal [3H]alanine uptake. The remaining proteoliposomes were diluted five times in external buffer supplemented with 500 nM unlabelled alanine and incubated for 30 min to allow efflux. Uptake during the 30 min incubation was assessed in parallel by 10 min preloading in 50 nM unlabeled alanine, followed by 5x dilution in external buffer containing 500 nM [3H]alanine ($7.95 \text{ Ci mmol}^{-1}$) and incubated for 30 min. All efflux experiments were performed at 30°C. Reactions were stopped by filtration on nitrocellulose filters and washed with ice cold stop buffer (200 mM KCl, 20 mM HEPES, pH 7.50). Filters were immersed into 3 ml scintillation cocktail solution, incubated for 24 hours on shaker and recorded on a 2450 MicroBeta microplate² counter.”

Discussion (page 17):

This paragraph has been revised to accommodate the new results and better explain our suggestive model the new paragraph is as follows:

“We propose that internal K^+ facilitates transport by binding to LeuT in a way that inhibits rebinding of Na^+ and, accordingly, also substrate. This will minimize substrate efflux and result in an increased concentrative capacity of the transporter, i.e. substrate can be transported against a larger chemical gradient. Unbinding of K^+ from LeuT will then favor an outward-open conformation of the transporter that in turn is further stabilized by Na^+ .”

Reviewer #3

Because there was uptake of [3H]-alanine by even in the absence of internal potassium-just like in SERT- it is quite likely that, also in LeuT, protons can functionally replace potassium. It is therefore very important to perform several of the experiments in the absence of potassium, but at lowered pH. This should not only include some of the transition metal ion FRET measurements, but also some uptake experiments in reconstituted proteoliposomes to see if stimulation of transport by an outward directed proton gradient (acidic inside) can be observed in the absence of potassium.

We also thank this Reviewer for positive and insightful comments. The reviewer points to a very important and relevant issue for the elucidation of the transport process. To address the question, we have performed a series of new experiments. As requested by the reviewer, we performed tmFRET experiments to assess the effect of pH on the transition to the outward-occluded conformation by K^+ binding. Unfortunately, we cannot obtain conclusive results from such experiments. As can be seen in Fig. 1 below, the experiment is not feasible because the lowered pH decreases the Ni^{2+} affinity for the His- X_3 -His site and accordingly, it is not possible to achieve a tmFRET plateau, which is critical for comparing different experimental conditions. Notably, such as decrease in Ni^{2+} affinity is expected by lowering pH because protons simply will compete for Ni^{2+} binding to the His- X_3 -His site (data in Fig. 1 are means \pm s.e.m. of three experiments performed in triplicates). Thus, we cannot address the effect of pH on conformational transitions by the tmFRET method.

We also investigated, as suggested by the reviewer, whether an outwardly directed proton gradient was able to stimulate of transport in the absence of K^+ . Proteoliposomes were formed in 200 mM Cs^+ pH 6 or pH 8 and [³H]alanine uptake was assessed at pH 8 in 200 mM Na^+ . As can be seen in Fig 2 below, the H^+ gradient appeared to have no effect on uptake (compare circles and tiles) (data are means \pm s.e.m. of three experiments performed in duplicates). This suggests that protons cannot substitute for K^+ as well as that a proton gradient does not stimulate uptake, at least not under the conditions used in the present investigation.

We should note that previously Na⁺/substrate symport-coupled H⁺ antiport was suggested for prokaryotic NSS proteins (Zomot et al. (2007) Nature; Zhao et al (2010) Nat Chem Biol) and for the homologous NSS protein Tyt1 it was shown that substrate transport elicits H⁺ efflux (Zhao et al (2010) Nat Chem Biol). The apparent absence in our experiments of an effect of a proton gradient warrants further investigations of the role of protons in transport mediated by LeuT; however, we find such experiments outside the scope of the present investigation that is focused on K⁺.

The data are now included in Results (page 6):

“Previously, Na⁺/substrate symport-coupled H⁺ antiport was suggested for prokaryotic NSS proteins^{39,40} and for the homologous NSS protein Tyt1 it was shown that substrate transport elicits H⁺ efflux⁴⁰. To test whether H⁺ could substitute for K⁺, proteoliposomes were formed in a CsCl buffer (200 mM) pH 6.5 or pH 8 and [³H]alanine uptake were assessed as in **Fig 1d** at pH 8. However, the H⁺ gradient appeared to have no effect on uptake: In proteoliposomes formed at pH 8, [³H]alanine uptake was 106 ± 6 % relative to uptake at pH 6.5 (means ± s.e.m., *n* = 3), suggesting that protons cannot substitute for K⁺ as well as that a proton gradient does not stimulate uptake, at least not under the conditions used in the present investigation.”

Minor issues:

P.3 line8: replace "several" by "all".

P8 last line from below: please enter the value for the EC50.

Fig. 1 Legend, line 12: "proteoliposomes containing..."/

Fig. 4 Legend, line 9 "purple" looks "pink" to me.

We have made all the minor changes as proposed.

Other minor changes not requested by the reviewers:

We reference in the revised version the newly published structure of the human SERT.

Reviewer #1 (Remarks to the Author)

The authors have adequately addressed my concerns.

Reviewer #2 (Remarks to the Author)

I have reviewed the revised manuscript and generally find that the authors's responses to my (and the other reviewers's criticisms reveal more uncertainty in the interpretations of the results than hinted at in the original version. The conclusions are weakened enough to raise questions about the significance of this work. Primarily, the issue of coupling of K to transport is of concern. The original premise of the paper is that the K binding provides a model for the similar coupling of K to transport in a eukaryotic NSS member. The authors now acknowledge that the effect of potassium is not due to strict coupling with transport, so the original premise no longer applies. Indeed, my own view of the data is that there is a good possibility that K is having its very low affinity effect via some kind of allosteric mechanism. The main interest of the work is through the analogy to mammalian transport, once that no longer applies, this work will be mainly of interest to a more specialized audience.

Reviewer #3 (Remarks to the Author)

Only one minor typo remains:

Abstract line 5: inc(r)eases

Reviewer #1

The authors have adequately addressed my concerns.

We thank the reviewer for all the constructive inputs.

Reviewer #2

I have reviewed the revised manuscript and generally find that the authors's responses to my (and the other reviewers's criticisms reveal more uncertainty in the interpretations of the results than hinted at in the original version. The conclusions are weakened enough to raise questions about the significance of this work. Primarily, the issue of coupling of K to transport is of concern. The original premise of the paper is that the K binding provides a model for the similar coupling of K to transport in a eukaryotic NSS member. The authors now acknowledge that the effect of potassium is not due to strict coupling with transport, so the original premise no longer applies. Indeed, my own view of the data is that there is a good possibility that K is having its very low affinity effect via some kind of allosteric mechanism. The main interest of the work is through the analogy to mammalian transport, once that no longer applies, this work will be mainly of interest to a more specialized audience.

We respectfully disagree with the reviewer that the conclusions are weakened by the new results presented in the manuscript. We do not suggest – and did not suggest in the first version of the manuscript – that the effect of K^+ is due to a *strict* coupling with transport. If the coupling was strict, we would not expect to observe any transport in the absence of K^+ , which is evident from Fig. 1e (not a new figure). However, the effect of K^+ on 5-HT uptake in SERT is not strictly coupled either: 5-HT uptake by SERT *in the absence* of K^+ is present in platelets (Nelson & Rudnick (1979) JBC), in placenta (Cool et al. (1990) Biochem), in HEK293 cells (Schicker et al (2012) JBC) and in oocytes (Adams & DeFelice (2003) BiophysJ). In fact, it has previously been suggested that the effect of K^+ on SERT is indeed through the interaction with a modulatory site (Adams & DeFelice (2003) BiophysJ). Thus, the original premises of the paper have *not* changed and indeed it is possible that the effect of K^+ on LeuT could be similar to the role of K^+ in 5-HT transport by SERT. Moreover, it is important to emphasize that, to our knowledge, there is no study on the K^+ affinity for SERT and thus it is not known whether this could be similar to that for LeuT. Finally, our paper does not only demonstrate a functional role of K^+ in uptake but also reveal important mechanistic insight into how K^+ can act on the transporter.

Taken together, we think that the original premises of the paper still apply and have even been strengthened by the new results.

Reviewer #3

Only one minor typo remains:

Abstract line 5: inc(r)eases.

We also thank this reviewer for all the constructive inputs in this process. The typo has been corrected.